# High levels of primary biogenic organic aerosols are driven by only a few plant-associated microbial taxa

Abdoulaye Samaké[1*], Aurélie Bonin[2], Jean-Luc Jaffrezo[1], Pierre Taberlet[2], Samuël Weber[1], Gaëlle Uzu[1], Véronique Jacob[1], Sébastien Conil[3], Jean M. F. Martins[1*]

[1]University Grenoble Alpes, CNRS, IRD, INP-G, IGE (UMR 5001), 38000 Grenoble, France
[2]University Grenoble Alpes, CNRS, LECA (UMR 5553), BP 53, 38041 Grenoble, France
[3]ANDRA DRD/OPE Observatoire Pérenne de l'Environnement, 55290 Bure, France

*Corresponding to:* Abdoulaye Samaké (abdoulaye.samake2@univ-grenoble-alpes.fr) and Jean Martins (jean.martins@univ-grenoble-alpes.fr)

**Abstract.** Primary biogenic organic aerosols (PBOA) represent a major fraction of coarse organic matter (OM) in air. Despite their implication in many atmospheric processes and human health problems, we surprisingly know little about PBOA characteristics (i.e., composition, dominant sources, and contribution to airborne-particles). In addition, specific primary sugar compounds (SCs) are generally used as markers of PBOA associated with bacteria and fungi but our knowledge of microbial communities associated with atmospheric particulate matter (PM) remains incomplete. This work aimed at providing a comprehensive understanding of the microbial fingerprints associated with SCs in $PM_{10}$ (particles smaller than 10μm) and their main sources in the surrounding environment (soils and vegetation). An intensive study was conducted on $PM_{10}$ collected at rural background site located in an agricultural area in France. We combined high-throughput sequencing of bacteria and fungi with detailed physicochemical characterization of $PM_{10}$, soils and plant samples, and monitored meteorology and agricultural activities throughout the sampling period. Results shows that in summer SCs in $PM_{10}$ are a major contributor of OM in air, representing 0.8 to 13.5% of OM mass. SCs concentrations are clearly determined by the abundance of only a few specific airborne fungi and bacteria taxa. The temporal fluctuations in the abundance of only 4 predominant fungal genera, namely *Cladosporium*, *Alternaria*, *Sporobolomyces* and *Dioszegia* reflect the temporal dynamics in SC concentrations. Among bacteria taxa, the abundance of only *Massilia*, *Pseudomonas*, *Frigoribacterium* and *Sphingomonas* are positively correlated with SC species. These microbial are significantly enhanced in leaf over soil samples. Interestingly, the overall community structure of bacteria and fungi are similar within $PM_{10}$ and leaf samples and significantly distinct between $PM_{10}$ and soil samples, indicating that surrounding vegetation are the major source of SC-associated microbial taxa in $PM_{10}$ on rural area of France.

## 1. Introduction

Airborne particulate matter (PM) is the subject of high scientific and political interests mainly because of its important effects on climate and public health (Boucher et al., 2013; Fröhlich-Nowoisky et al., 2016; Fuzzi et al., 2006). Numerous epidemiological studies have significantly related both acute and chronic exposures to ambient PM with respiratory impairments, heart diseases, asthma, lung cancer, as well as increased risk of mortality (Kelly and Fussell, 2015; Pope and Dockery, 2006). PM can also affect directly or indirectly the climate by absorbing and/or diffusing both the incoming and outgoing solar radiation (Boucher et al., 2013; Fröhlich-Nowoisky et al., 2016). These effects are modulated by highly variable physical characteristics (e.g., size, specific surface, concentrations, etc.) and complex chemical composition of PM (Fröhlich-Nowoisky et al., 2016; Fuzzi et al., 2015). PM consists of a complex mixture of inorganic, trace elements and carbonaceous matter (organic carbon and elemental carbon) with organic matter (OM) being generally the major but poorly characterized constituent of PM (Boucher et al., 2013; Bozzetti et al., 2016). A quantitative understanding of OM sources is critically important to develop efficient guidelines for both air quality control and abatement strategies. So far, considerable efforts have been undertaken to investigate OM associated with anthropogenic and secondary sources, but much less is known about emissions from primary biogenic sources (Bozzetti et al., 2016; China et al., 2018; Yan et al., 2019).

Primary biogenic organic aerosols (PBOAs) are a subset of organic PM that are directly emitted by processes involving the biosphere (Boucher et al., 2013; Elbert et al., 2007). PBOAs refer typically to biologically derived materials, notably including living organisms (bacteria, fungal spores, Protozoa, viruses) and non-living biomass (e.g., microbial fragments) and other types of biological materials like pollen or plant debris (Amato et al., 2017; Elbert et al., 2007; Fröhlich-Nowoisky et al., 2016). PBOAs are gaining increasing attention notably because of their ability to affect human health by causing infectious, toxic, and hypersensitivity diseases (Fröhlich-Nowoisky et al., 2016; Huffman et al., 2019). For instance, PBOA components, especially fungal spores and bacterial cells, have recently been shown to cause significant oxidative potential (Samaké et al., 2017). However, to date, the precise role of PBOA components and interplay regarding mechanisms of diseases are remarkably misunderstood (Coz et al., 2010; Hill et al., 2017). Specific PBOA components can also participate in many relevant atmospheric processes like cloud condensation and ice nucleation, thereby directly or indirectly affecting the Earth's hydrological cycle and radiative balance (Boucher et al., 2013; Fröhlich-Nowoisky et al., 2016; Hill et al., 2017). These diverse impacts are effective at a regional scale due to the transport of PBOAs (Dommergue et al., 2019; Yu et al., 2016). Moreover, PBOAs are a major component of OM found in particles less than 10 μm in aerodynamic diameter ($PM_{10}$) (Bozzetti et al., 2016; Coz et al., 2010; Samaké et al., 2019b). For instance, Bozzetti et al. (2016) have shown that PBOAs equal the contribution of secondary organic aerosols (SOAs) to OM in $PM_{10}$ collected at a rural background site in Switzerland during both the summer and winter periods. However, current estimates of global terrestrial PBOA emissions are very uncertain and range between 50 and 1000 Tg y$^{-1}$ (Boucher

et al., 2013; Coz et al., 2010; Elbert et al., 2007), underlining the critical gap in the understanding of this significant OM fraction.

The recent application of fluorescent technics such as ultraviolet aerodynamic particle sizer, wideband integrated bioaerosol sensor (Bozzetti et al., 2016; Gosselin et al., 2016; Huffman and Santarpia, 2017; Huffman et al., 2019), or scanning electron microscopy (Coz et al., 2010) have provided very insightful information on the abundance of size segregated ambient PBOAs. Atmospheric sources of PBOAs are numerous and include agricultural activities, leaf abrasion, and soil resuspension. (Coz et al., 2010; Medeiros et al., 2006; Pietrogrande et al., 2014). To date, the detailed constituents of PBOAs, their predominant sources and atmospheric emission processes as well as their contributions to total airborne particles remain poorly documented and quantified (Bozzetti et al., 2016; Coz et al., 2010; Elbert et al., 2007). Such information would be important for investigating the properties and atmospheric impacts of PBOAs as well as for a future optimization of source-resolved chemical transport models (CTM), which are still generally unable to accurately simulate important OM fractions (Ciarelli et al., 2016; Heald et al., 2011; Kang et al., 2018).

Primary sugar compounds (SC, defined as sugar alcohols and saccharides) are ubiquitous water-soluble compounds found in atmospheric PM (Gosselin et al., 2016; Medeiros et al., 2006; Pietrogrande et al., 2014; Jia et al., 2010b). SC species are emitted from biologically derived sources (Medeiros et al., 2006, Verma et al., 2018) and have sometimes been detected in aerosols taken from air masses influenced by smoke from biomass burning (Fu et al., 2012; Yang et al., 2012). However, recent studies conducted at several sites across France revealed a weak correlation between daily concentrations of SC and levoglucosan in $PM_{2.5}$ and $PM_{10}$ collected throughout the year (Golly et al., 2018; Samaké et al., 2019a). This suggests that open burning of biomass is not a significant source of SC in the environments studied here. In this context, specific SC species are still extensively viewed as powerful markers for tracking sources and estimating PBOA contributions to OM mass (Bauer et al., 2008; Gosselin et al., 2016; Jia et al., 2010b; Medeiros et al., 2006). For example, glucose is the most common monosaccharide in vascular plants and it has been predominantly used as indicator of plant material (such as pollen or plant debris) from several areas around the world (Jia et al., 2010b; Medeiros et al., 2006; Pietrogrande et al., 2014; Verma et al., 2018). Trehalose (aka mycose) is a common metabolite of various microorganisms, serving as an osmoprotectant accumulating in cells cytosol during harsh conditions (e.g., dehydration and heat) (Bougouffa et al., 2014). It has been proposed as a generic indicator of soil-borne microbiota (Jia et al., 2010b; Medeiros et al., 2006; Pietrogrande et al., 2014; Verma et al., 2018). Similarly, mannitol and arabitol are two very common sugar alcohols (also called polyols) serving as storage and transport solutes in fungi (Gosselin et al., 2016; Medeiros et al., 2006; Verma et al., 2018). Their atmospheric concentrations levels have frequently been used to investigate fungal spores contributions to PBOAs mass in different environments (urban, rural, costal, and polar) around the world (Barbaro et al., 2015; Gosselin et al., 2016; Jia et al., 2010b; Verma et al., 2018; Weber et al., 2018).

Despite the relatively vast literature using the atmospheric concentration levels of SC as potential suitable markers of PBOAs associated with bacteria and fungi, our understanding of associated airborne microbial communities (i.e., diversity and community composition) remains poor. This is due in particular to the lack of high-resolution (i.e., daily) data sets characterizing how well the variability of these microbial communities may be related to that of primary sugar species. Such information is of paramount importance to better understand the dominant atmospheric sources of SC (and then PBOAs) as well as their relevant effective environmental drivers, which are still poorly documented (Bozzetti et al., 2016).

Our recent works discussed the size distribution features as well as the spatial and temporal variability in atmospheric particulate SC concentrations in France (Golly et al., 2018; Samaké et al., 2019a, 2019b). As a continuation, in this study, we present the first daily temporal concurrent characterization of ambient SC species concentrations and both bacterial and fungal community compositions for $PM_{10}$ collected at a rural background site located in an intensive agricultural area. The aim of this study was to use a DNA metabarcoding approach (Taberlet et al., 2018) to investigate $PM_{10}$-associated microbial communities, which can help answering the following research questions: (i) What are the microbial community structures associated with $PM_{10}$? (ii) Is the temporal dynamics of SC concentrations related to changes of the airborne microbial community compositions? (iii) What are the predominant sources of SC-associated microbial communities at a continental rural field site? Since soil and vegetation are currently believed to be the dominant sources of airborne microorganisms in most continental areas (Bowers et al., 2011; Jia et al., 2010a; Rathnayake et al., 2016), our study focused on these two potential sources.

## 2. Material and methods

### 2.1. Site description

The Observatoire Pérenne de l'Environnement (OPE) is a continental rural background observatory located at about 230 km east of Paris at an altitude of 392 m (Fig. 1). This French Critical Zone Observatory (CZO) is part of a long term multi-disciplinary project monitoring the state of environmental variables including among other fluxes, abiotic and biotic variables, and their functions and dynamics (http://ope.andra.fr/index.php?lang=en, last access: December 10, 2019). It is largely impacted by agricultural activities. It is also characterized by a low population density (less than 22 per km² within an area of 900 km$^2$), with no industrial activities nor surrounding major transport road. The air monitoring site itself lies in a "*reference sector*" of 240 km$^2$, in the middle of a field crop area (tens of kilometers in all directions). This reference sector is composed of vast farmlands interspersed with wooded areas. The area is further defined by a homogeneous soil type, with a predominantly superficial clay-limestone composition. The daily agricultural practices and meteorological data (including wind speed and direction, temperature, rainfall level and relative humidity) within the reference sector are recorded and made available by ANDRA (Agence nationale pour la gestion des déchets radioactifs). The agricultural fields of the area are generally submitted to a 3-year crop-rotation system. The major crops during the campaign period were pea and oilseed rape.

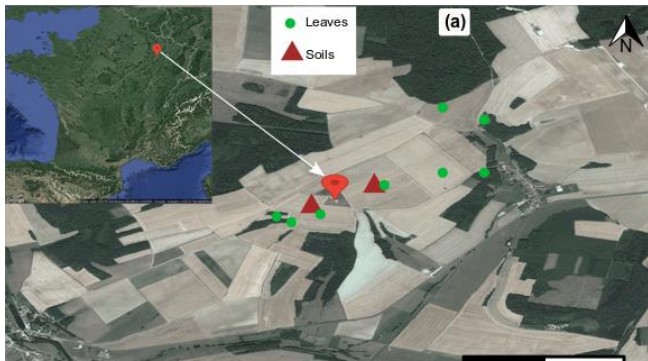 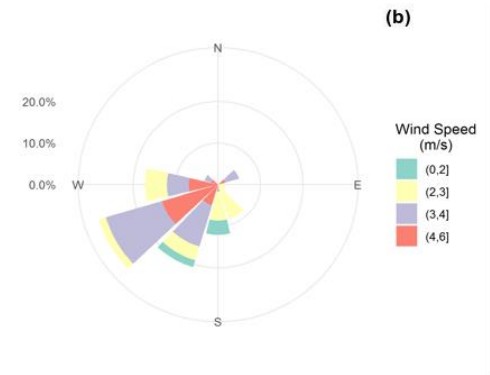

**Figure 1 : Overview of the sampling area at the OPE site (France). (A) Location of sampling units and (B) wind conditions (speed and direction) during the field sampling campaign period.**

### 2.2. Samples collection

An intensive field campaign was conducted at this site for the sake of the present study. The aerosol sampling campaign period lasted from June 12th to August 21st, 2017, covering the summer period in France. During this period, ambient PM$_{10}$ were collected daily (starting at 9 am UTC to 9 am UTC the next day) onto prebaked quartz fiber filters (Tissuquartz PALL QAT-UP 2500, Ø = 150 mm) using high volume samplers (Aerosol Sampler DHA-80, Digitel; 24 h at 30 m$^3$ h$^{-1}$). After collection, all filter samples were wrapped in aluminum foils, sealed in zipper plastic bags, and stored at < 4 °C until further analysis. More details on the preparation, storage, and handling of these filter samples can be found in Samaké et al. (2019b). A total of 69 samples and six field blanks were collected.

Surface soil samples (0-5 cm depth, 15x15 cm area) were simultaneously collected from two fields, within pea and oilseed rape-growing areas, respectively. The fields are located in the immediate vicinity of the PM$_{10}$ sampler and under the prevailing wind directions (Fig. 1). To represent as closely as possible the local soil microbial communities, we randomly collected five subsamples (about 100g per sampling unit) within each parcel and pooled them. Topsoil sampling took place on a weekly basis along the campaign period. After collection and homogenization, 15g of each subsample were stored in airtight containers (sterile bottles, Schott, GL45, 100ml) containing the same weight of sterile silica gel (around 15g). Such soil desiccation method is a straightforward approach to prevent any microbial growth and change in community over time at room temperature (Taberlet et al., 2018). A total of eight topsoil samples were collected for each parcel.

Finally, leaf samples were collected from the major types of vegetables within the reference sector. These include leaf of oilseed rape, pea, oak, maples, beech, and herbs (Fig. 1). A total of eight leaf samples were analyzed. These samples were also stored in airtight containers (sterile bottles, Schott, GL45, 100ml) containing 15g of silica gel.

It should be noted that leaf samples were collected only once, four weeks after the end of PM and soil sampling,
while the major crops were still on site.

## 2.3. Chemical analyses

Daily $PM_{10}$ samples were analyzed for various chemical species using subsampled fractions of the collection filters
and a large array of analytical methods. Detailed information on all the chemical analysis procedures have been
reported previously (Golly et al., 2018; Samaké et al., 2019b; Waked et al., 2014). Briefly, SCs (i.e. polyols and
saccharides) and water-soluble ions (including $Ca^{2+}$) have been systematically analyzed in all samples, using
respectively high-performance liquid chromatography with pulsed amperometric detection (HPLC-PAD) and
ionic chromatography (IC, Thermo Fisher ICS 3000, USA). Free-cellulose concentrations were determined using
an optimized enzymatic hydrolysis (Samaké et al., 2019a) and the subsequent analysis method of the resultant
glucose units with an HPLC-PAD (Golly et al., 2018; Samaké et al., 2019b; Waked et al., 2014). Organic and
elemental carbon (OC, EC) have been analyzed using a Sunset thermo-optic instrument and the EUSAAR2
protocol (Cavalli et al., 2010). This analytical method requires high temperature, thereby constraining the choice
of quartz as sampling filter material. OM content in $PM_{10}$ samples were then estimated using an OM-to-OC
conversion factor of 1.8: $OM = 1.8 \times OC$ (Samaké et al., 2019b, 2019a). This value of 1.8 for the OM/OC ratio
was chosen on the basis of previous studies carried out in France (Samaké et al., 2019b, and reference therein)

## 2.4. Biological analyses: DNA extraction in $PM_{10}$ samples

Aerosol samples typically contain very low DNA concentrations, and the DNA-binding properties of quartz fibers
of aerosol collection filters make challenging its extraction with traditional protocols (Dommergue et al., 2019;
Jiang et al., 2015; Luhung et al., 2015). In the present study, we were also constrained by the limited available
daily collection filter surface for simultaneous chemical and microbiological analyses of the same filters. To
circumvent issues of low efficiency during genomic DNA extraction, several technical improvements have been
made to optimize the extraction of high-quality DNA from $PM_{10}$ samples (Dommergue et al., 2019; Jiang et al.,
2015; Luhung et al., 2015). These include thermal water bath sonication helping lysis of thick cell walls (e.g.,
fungal spores and gram-positive bacteria), which might not be effectively lysed by means of sole bead beating
(Luhung et al., 2015). Some consecutive (2 days at maximum) quartz filter samples with low OM concentrations
were also pooled when necessary. Detailed information regarding the resultant composite samples (labeled as A1
to A36) are presented in Table S1. Figure S1 presents the average concentration levels of SC species in each
sample. The results clearly show that air samples can be categorized from low (background, from A1 to A4 and
A21 to A36) to high (peak, from A5 to A20) $PM_{10}$ SC concentration levels.
In terms of DNA extraction, ¼ (about 38.5 $cm^2$) of each filter sample were used. First, filter aliquots were
aseptically inserted into individual 50 mL Falcon tubes filed with sterilized saturated phosphate buffer ($Na_2HPO4$,
$NaH_2PO_4$, 0.12 M; pH ≈ 8). $PM_{10}$ were desorbed from the filter samples by gentle shaking for 10 min at 250 rpm.
This pretreatment allows the separation of the collected particles from quartz filters thanks to the high competing
interaction between saturated phosphate buffer and charged biological materials (Jiang et al., 2015; Taberlet et al.,
2018). After gentle vortex mixing, the subsequent resuspension was filtered with a polyethersulphone membrane
disc filter (PES, Supor® 47mm 200, 0.2 µm, PALL, USA). We repeated this desorbing step three times to enhance
the recovery of biological material from quartz filters. Each collection PES membrane was then shred into small
pieces and used for DNA extractions using the DNeasy PowerWater kit (Qiagen, Germantown, MD, USA). The
standard protocol of the supplier was followed, with only minor modifications: 30 min of thermal water bath
sonication at 65°C (EMAG, Emmi-60 HC, Germany; 50% of efficiency), and 5 min of bead beating before and
after sonication were added. Finally, DNA was eluted in 50 µl of EB buffer. Such an optimized protocol has been
recently shown to produce a 10-fold increase in DNA extraction efficiency (Dommergue et al., 2019; Luhung et
al., 2015), thereby allowing high-throughput sequencing of air samples. Note that all the steps mentioned above
were performed under laminar flow hoods, and that materials (filter funnels, forceps, and scissors) were sterilized
prior to use.

## 2.4.1. Biological analyses: DNA extraction from soil and leaf samples

The soil samples pretreatment and extracellular DNA extraction were achieved following an optimized protocol
proposed elsewhere (Taberlet et al., 2018). Briefly, this protocol involves mixing thoroughly and extracting 15g
of soil in 15 ml of sterile saturated phosphate buffer for 15 min. About 2 mL of the resulting extracts were
centrifuged for 10 min at 10,000g, and 500 μL of the resulting supernatant were used for DNA extraction using
the NucleoSpin Soil Kit (Macherey-Nagel, Düren, Germany) following the manufacturer's original protocol after
skipping the cells lysis step. Finally, DNA was eluted with 100 μL of SE buffer.
To extract DNA from either endophytic or epiphytic microorganisms, aliquots of leaf samples (about 25–30mg)
were extracted with the DNeasy Plant Mini Kit (QIAGEN, Germany) according to the supplier's instructions, with
the following minor modifications: after the resuspension of powdered samples in 400 μL of AP1 buffer, the
samples were incubated for 45 min at 65°C with RNase A. Finally, DNA was eluted with 100 μL of AE buffer.

### 2.4.2. Biological analyses: PCR amplification and sequencing

Bacterial and fungal community compositions were surveyed using respectively the Bact02 (Forward 5'—
KGCCAGCMGCCGCGGTAA—3' and Reverse 3'—GGACTACCMGGGTATCTAA—5') and Fung02
(Forward          5'—GGAAGTAAAAGTCGTAACAAGG—3'          and          Reverse          3'—
CAAGAGATCCGTTGYTGAAAGTK—5') published primer pairs [see (Taberlet et al., 2018) for details on
these primers]. The primer pair Bact02 targets the V4 region of the bacterial 16S rDNA region while the Fung02
primer pair targets the nuclear ribosomal internal transcribed spacer region 1 (ITS1). Four independent PCR
replicates were carried out for each DNA extract. Eight-nucleotide tags were added to both primer ends to uniquely
identify each sample, ensuring that each PCR replicate is labeled by a unique combination of forward and reverse
tags. The tag sequence were created with the *oligotag* command within the open-source OBITools software suite
(Boyer et al., 2016), so that all pairwise tag combinations were differentiated by at least five different base pairs
(Taberlet et al., 2018).
DNA amplification was performed in a 20-μL total volume containing 10 μL of AmpliTaq Gold 360 Master Mix
(Applied Biosystems, Foster City, CA, USA), 0.16 μL of 20 mg ml-1 bovine serum albumin (BSA; Roche
Diagnostics, Basel, Switzerland), 0.2 μM of each primer, and 2 μL of diluted DNA extract. DNA extracts from
soil and filters were diluted eight times while DNA extracts from leaves were diluted four times. Amplifications
were performed using the following thermocycling program: an initial activation of DNA polymerase for 10 min
at 95°C; x cycles of 30 s denaturation at 95°C, 30 s annealing at 53°C and 56°C for bacteria and fungi, respectively,
90 s elongation at 72°C; and a final extension at 72°C for 7 min. The number of cycles x was determined by qPCR
and set at 40 for all markers and DNA extract types, except for the Bact02 amplification of soil and leaf samples
(30 cycles), and the Fung02 amplification of filter samples (42 cycles). After amplification, about 10% of
amplification products were randomly selected and verified using a QIAxel Advance device (QIAGEN, Hilden,
Germany) equipped with a high-resolution cartridge for separation.
After amplification, PCR products from the same marker were pooled in equal volumes and cleaned with the
MinElute PCR purification kit (Qiagen, Hilden, Germany) following the manufacturer's instructions. The two
pools were sent to Fasteris SA (Geneva, Switzerland; https://www.fasteris.com/dna/; last access December 10,
2019) for library preparation and MiSeq Illumina 2×250 bp paired-end sequencing. The two sequencing libraries
(one per marker) were prepared according to the PCR-free MetaFast protocol (www.fasteris.com/metafast, last
access December 10, 2019), which aims at limiting the formation of chimeras.
To monitor any potential false positives inherent to tag jumps and contaminations (Schnell et al., 2015), sequencing
experiment included both extraction and PCR negatives, as well as unused tag combinations.

### 2.4.3. Bio-informatic analyses of raw reads

The Illumina raw sequence reads were processed separately for each library using the OBITools software suite
(Boyer et al., 2016), specifically dedicated to metabarcoding data processing. First, the raw paired-ends were
assembled using the *illuminapairedend* program, and the sequences with a low alignment score (fastq average
quality score < 40) were discarded. The aligned sequences were then assigned to the corresponding PCR replicates
with the program *ngsfilter,* by allowing zero and two mismatches on tags and primers, respectively. Strictly
identical sequences were dereplicated using the program *obuniq*, and a basic filtration step was performed with
the *obigrep* program to select sequences within the expected range length (i.e., longer than 65 or 39 bp for fungi
and bacteria, respectively, excluding tags and primers), without ambiguous nucleotides, and observed at least 10
times in at least one PCR replicate.
The remaining unique sequences were grouped and assigned to Molecular Taxonomic Units (MOTUs) with a 97%
sequence identity using the *Sumatra* and *Sumaclust* programs (Mercier et al., 2013). The *Sumatra* algorithm
computes pairwise similarities among sequences based on the length of the Longest Common Subsequence and
the *Sumaclust* program uses these similarities to cluster the sequences (Mercier et al., 2013). Abundance of
sequences belonging to the same cluster were summed up and the cluster center was defined as the MOTU
representative of the cluster (Mercier et al., 2013).
The taxonomic classification of each MOTU was performed using the *ecotag* program (Boyer et al., 2016), which
uses full-length metabarcodes as references. The *ecoPCR* program (Ficetola et al., 2010) was used to build the
metabarcode reference database for each marker. Briefly, *ecoPCR* performs an *in silico* amplification within the
EMBL public database (release 133) using the Fung02 and Bact02 primer pairs and allowing a maximum of three
mismatches per primer. The resultant reference database was further refined by keeping only sequence records
assigned at the species, genus and family levels.
After taxonomic assignment datasets were acquired, further processing with the open source R software (R studio
interface, version 3.4.1) was performed to filter out chimeras, potential contaminants, chimeras and failed PCR
replicates. More specifically, MOTUs that were highly dissimilar to any reference sequence (sequence identity <
0.95) were considered as chimeras and discarded. Secondly, MOTUs whose abundance was higher in extraction
or PCR negatives were also excluded. Finally, PCR replicates inconstantly distant from the barycenter of the four
PCR replicates corresponding to the same sample were considered as dysfunctional and discarded. The remaining
PCR replicates were summed up per sample.

## 2.5. Data analysis

Unless specified otherwise, all exploratory statistical analyses were achieved with R. Rarefaction and extrapolation
curves were obtained with the *iNext* 2.0-12 package (Hsieh et al., 2016), to investigate the gain in species richness
as we increased the sequencing depth for each sample. Alpha diversity estimators including Shannon and Chao1
were calculated with the *phyloseq* 1.22-3 package (McMurdie and Holmes, 2013), on data rarefied to the same
sequencing depth per sample type (see Table S2 for details on the rarefaction depths). Non-metric
multidimensional scaling (NMDS) ordination analysis was performed to decipher the temporal patterns in airborne
microbial community structures (phylum or class taxonomic group) in air samples. These analyses were achieved
with the *metaMDS* function within the *vegan* package (Oksanen et al., 2019) with the number random starts set to
500. The NMDS ordinations were obtained using pairwise dissimilarity matrices based on Bray Curtis index. The
*envfit* function implemented in vegan was used to assess the airborne microbial communities that could explain
the temporal dynamics of ambient SC species concentrations. Pairwise analysis of similarity (ANOSIM) was
performed to assess similarity between groups of $PM_{10}$ aerosols sample. This was achieved using the *anosim*
function of *vegan* (Oksanen et al., 2019), with the number of permutations sets to 999. Spearman's rank correlation
analysis was used to investigate further the relationship between airborne microbial communities and SC species.
To gain further insight into the dominant source of SC-associated microbial communities, NMDS analysis based
on Horn distance was performed to compare the microbial community composition similarities between $PM_{10}$
aerosols, soils, and leaf samples.

## 3.  Results

### 3.1. Primary sugar compounds (SC), and relative contributions to OM mass

Temporal dynamics of daily $PM_{10}$ carbonaceous components (e.g., primary sugar compounds, cellulose and OM)
are presented in Fig. 2. Nine SCs including seven polyols and two saccharide compounds have been quantified in
all ambient $PM_{10}$ collected at the study site. Ambient SC concentration levels peaked on August $8^{th}$, 2017, in
excellent agreement with the daily harvest activities around the study site (Fig. 2A). The average concentrations
(average ± SD) of total SCs during the campaign are 259.8 ± 253.8 ng m$^{-3}$, with a range of 26.6 to 1 679.5 ng m$^{-3}$,
contributing on average to 5.7 ± 3.2% of total OM mass in $PM_{10}$, with a range of 0.8–13.5% (Fig. 2B). The total
measured polyols present average concentrations of 26.3 ± 54.4 ng m$^{-3}$. Among all the measured polyols, arabitol
(67.4 ± 83.1 ng m$^{-3}$) and mannitol (68.1 ± 75.3 ng m$^{-3}$) are the predominant species, followed by lesser amounts
of sorbitol (10.9 ± 7.6 ng m$^{-3}$), erythritol (7.0 ± 8.8 ng m$^{-3}$), inositol (2.3 ± 2.0 ng m$^{-3}$), and xylitol (2.3 ± 3.0 ng m$^-$

[3]). Glycerol was also observed in our samples, but with concentrations frequently below the quantification limit. The average concentrations of saccharide compounds are $51.2 \pm 45.0$ ng m$^{-3}$. Threalose ($55.8 \pm 51.9$ ng m$^{-3}$) is the most abundant saccharide species, followed by glucose ($46.9 \pm 37.1$ ng m$^{-3}$). The average concentrations of calcium are $251.1 \pm 248.4$ ng m$^{-3}$.

A Spearman's rank correlation analysis based on the daily dynamics was used to examine the relationships between SC species. As shown in Table 1, sorbitol and inositol are well linearly correlated ($R = 0.57$, $p < 0.001$). Herein, sorbitol ($R = 0.59$, $p < 0.001$) and inositol ($R = 0.64$, $p < 0.001$) are significantly correlated to Ca$^{2+}$. It can also be noted that all other SC species are highly correlated with each other ($p < 0.001$) and that they are weakly correlated to the temporal dynamics of sorbitol and inositol (Table 1).

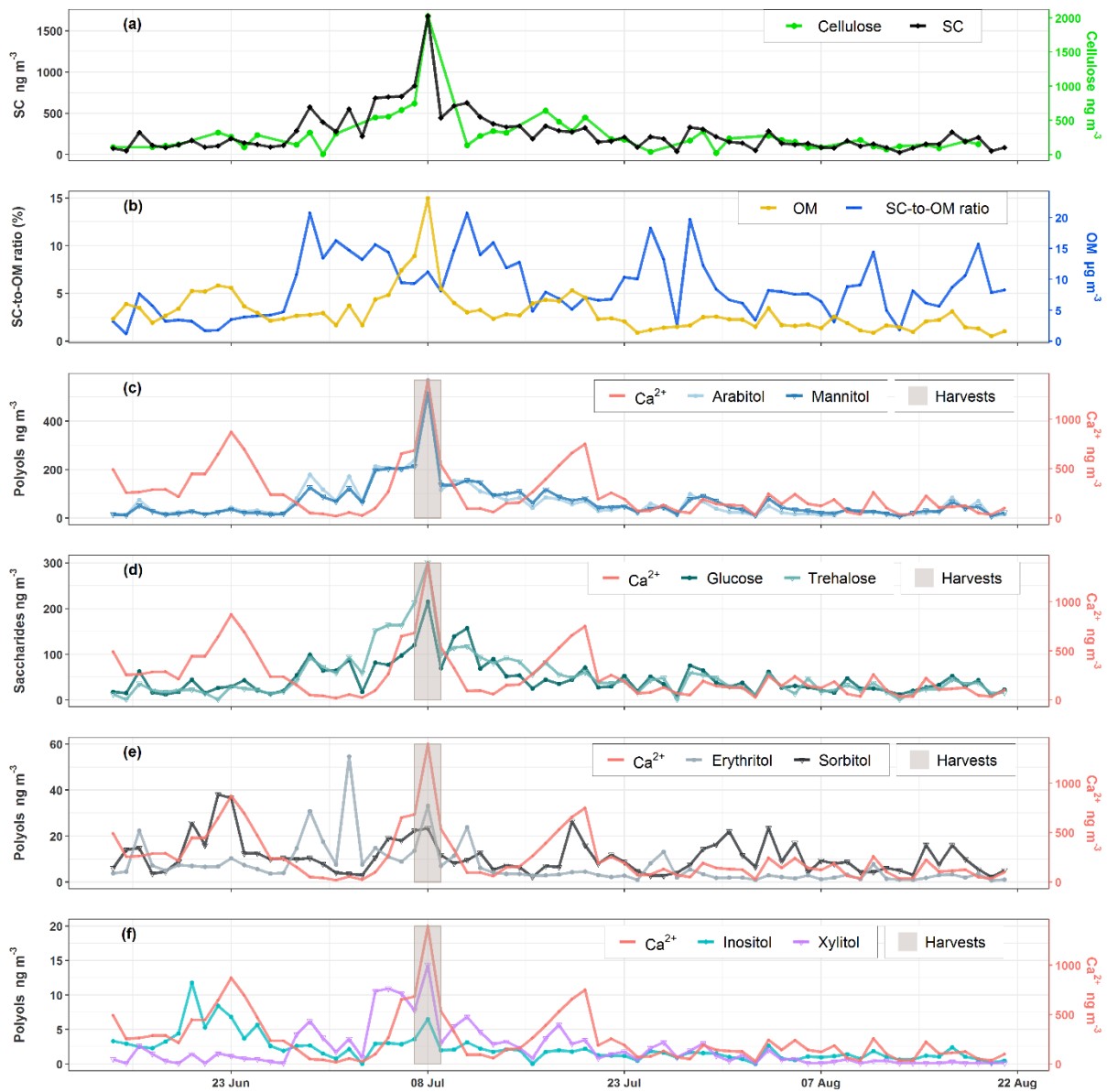

**Figure 2: Ambient concentrations of carbonaceous components in PM$_{10}$. (A; C to F) Daily variations of SCs and calcium concentrations along with daily agricultural activities around the site. (B) Contribution of SCs to organic matter mass. Results for nine-week daily measurements indicate that SCs together represent a large fraction of OM, contributing between 0.8 to 13.5% to OM mass in summer. Glycerol is not presented because its concentration was generally below the quantification limit.**

**Table 1 : Relationships between SCs and calcium in PM$_{10}$ from the study site. Spearman's rank correlation analyses**
**are based on the daily dynamics of chemicals species (n= 69).**

|  | Arabitol | Mannitol | Glucose | Trehalose | Erythritol | Xylitol | Sorbitol | Inositol | Ca$^{2+}$ |
|---|---|---|---|---|---|---|---|---|---|
| Arabitol | 1.00 | | | | | | | | |
| Mannitol | 0.94*** | 1.00 | | | | | | | |
| Glucose | 0.90*** | 0.90*** | 1.00 | | | | | | |
| Trehalose | 0.93*** | 0.96*** | 0.87*** | 1.00 | | | | | |
| Erythritol | 0.69*** | 0.51*** | 0.57*** | 0.56*** | 1.00 | | | | |
| Xylitol | 0.84*** | 0.84*** | 0.80*** | 0.79*** | 0.65*** | 1.00 | | | |
| Sorbitol | 0.22 | 0.26* | 0.35** | 0.15 | 0.21 | 0.24* | 1.00 | | |
| Inositol | 0.39** | 0.24 | 0.34** | 0.25* | 0.71*** | 0.39** | 0.57*** | 1.00 | |
| Ca$^{2+}$ | 0.12 | 0.11 | 0.11 | 0.09 | 0.30* | 0.27* | 0.59*** | 0.64*** | 1.00 |
| Note | * p < 0.1 | ** p < 0.01 | *** p < 0.001 | | | | | | |

**3.2. Microbial characterization of samples, richness and diversity**

The structures of bacterial and fungal communities were generated for the 62 collected samples, consisting of 36 aerosol, 18 surface soil, and 8 leaf samples. After paired-end assembly of sequence reads, sample assignment, filtering based on sequence length and quality and discarding of rare sequences, we are left with 2,575,857 and 1,647,000 reads respectively for fungi and bacteria, corresponding respectively to 4,762 and 5,852 unique sequences, respectively. After the clustering of high-quality sequences, potential contaminants and chimeras, the final data sets (all samples pooled) consist respectively of 597 and 944 MOTUs for fungi and bacteria, with 1,959,549 and 901,539 reads. The average numbers of reads (average ± SE) per sample are 31,607 ± 2,072 and 14,563 ± 1,221, respectively. The rarefaction curves of MOTU diversity showed common logarithmic shapes approaching a plateau in all cases (Fig. S2). This indicates an overall sufficient sequencing depth to capture the diversity of sequences occurring in the different types of samples. To compare the microbial community diversity and species richness, data normalization was performed out by selecting randomly from each sample 4,287 fungal sequences and 2,865 bacterial sequence reads. The Chao1-values of fungi are higher for aerosol samples than for soil and leaf samples (p< 0.05), indicating higher richness in airborne PM$_{10}$ (Fig. S3A). In contrast, PM$_{10}$ and soil samples showed higher values of Shannon index (p< 0.05), indicating a higher fungal diversity in these ecosystems. The soil harbors higher bacterial richness and diversity than PM$_{10}$ (p< 0.05), which in turns harbors greater richness and diversity compared to leaf samples (p< 0.05) (Fig. S3B).

**3.3. Taxonomic composition of airborne PM$_{10}$**

**3.3.1. Fungal communities**

Statistical assignment of airborne PM$_{10}$ fungal MOTUs at different taxonomic levels reveals 3 phyla, 17 classes, 58 orders and 160 families (Fig. 3). Interestingly, fungal MOTUs are dominated by two common phyla: Ascomycota (accounting for an average of 76 ± 20.4% (average ± SD)) of fungal sequences across all air samples, followed by Basidiomycota (23.9± 20.4%). The remaining sequences correspond to Mucoromycota (< 0.01%) and to unclassified sequences (approximately 0.03%). As evidenced in Fig. 3, the predominant (> 1%) fungal classes are Dothideomycetes (70.0%), followed by Agaricomycetes (16.0%), Tremellomycetes (5.0%), Sordariomycetes (2.6%), Microbotryomycetes (2.2%), Leotiomycetes (1.8%) and Eurotiomycetes (1.4%). The predominant orders include Pleosporales (35.5 %) and Capnodiales (34.4 %), which belong to Ascomycota. Likewise, the dominant orders in Basidiomycota are Polyporales (7.5%), followed by Russulales (4.2%), Tremellales (2.8%), Hymenochaetales (2.6%) and Sporidiobolales (2.2%). At the genus level, about 327 taxa are characterized across all air samples, among which *Cladosporium* (32.9%), *Alternaria* (15.0%), *Epicoccum* (15.0%), *Peniophora* (2.7%), *Sporobolomyces* (2.2 %), *Phlebia* (2.0%) and *Pyrenophora* (1.9 %) are the most abundant communities.

**3.3.2. Bacterial communities**

For bacterial communities, the Bact02 marker allowed identifying 17 phyla, 43 classes, 91 orders and 182 families (Fig. 3). Predominant phyla include Proteobacteria (55.3±8.6%), followed by Bacteroidetes (22.1±4.1%), Actinobacteria (14.2±2.2%), Firmicutes (6±5.9%), with less than 1.8 % of the total bacterial sequence reads being unclassified. At the class level, the predominant bacteria are Alphaproteobacteria (29.4%), Actinobacteria (13.8%), Gammaproteobacteria (12.1%), Betaproteobacteria (11.4%), Cytophagia (8.3%), Flavobacteriia (6.3%),

Sphingobacteriia (5.9%), Bacilli (3.5%) and Clostridia (2.2%). As many as 392 genera were detected in all aerosol samples, although many sequences (22.8%) could not be taxonomically assigned at the genus level. The most abundant (> 2%) genera are *Sphingomonas* (20.0%), followed by *Massilia* (8.4%), *Hymenobacter* (5.5%), *Pseudomonas* (5.1%), *Pedobacter* (3.3%), *Flavobacterium* (2.8%), *Chryseobacterium* (2.8%), *Frigoribacterium* (2.5%), and *Methylobacterium* (1.9%).

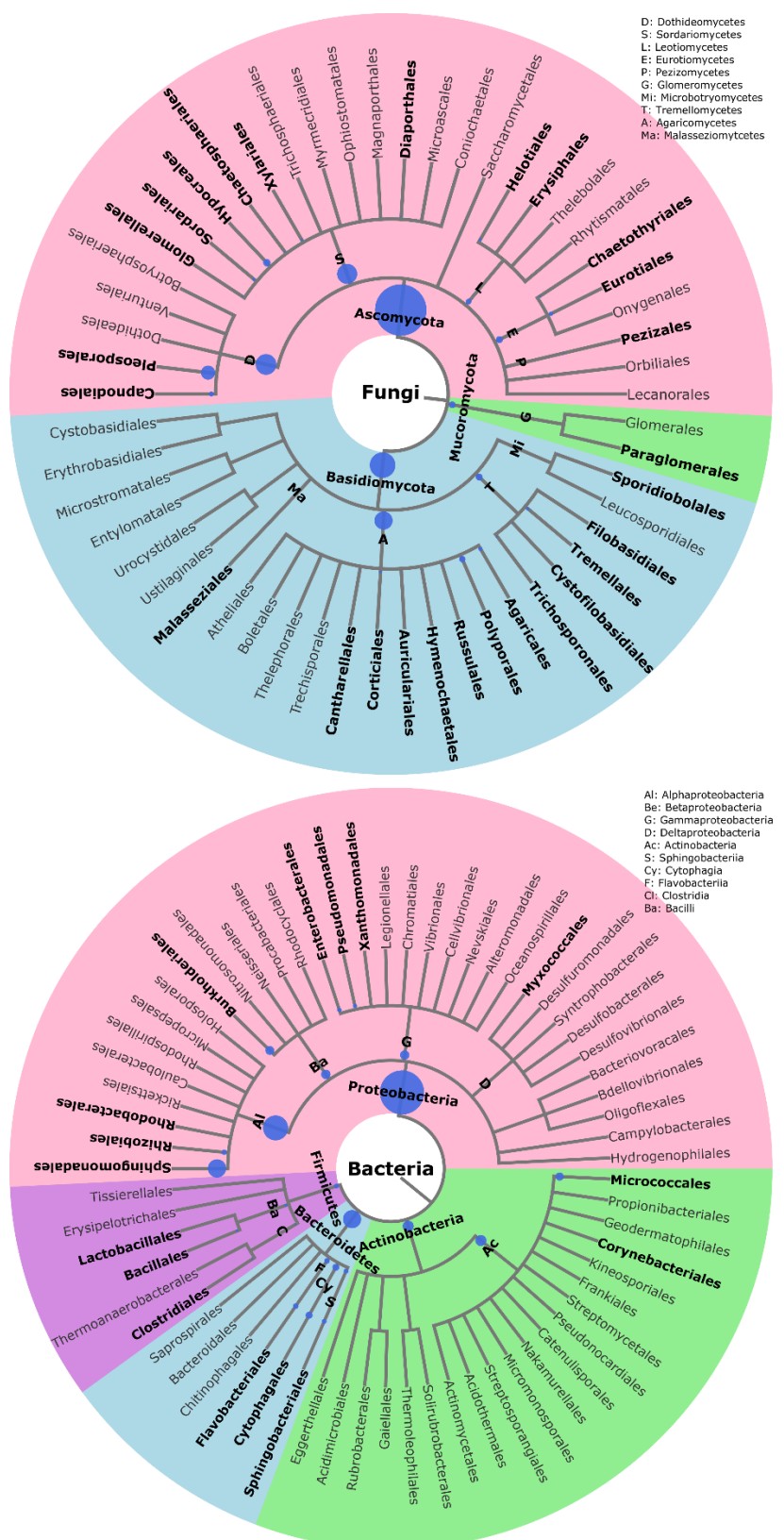

356

Figure 3: Taxonomic and phylogenetic trees of fungal and bacterial community structure in PM$_{10}$ at the study site.
357  Phylogenetic trees are analysed with the Environment for Tree Exploration (*ETE3*) package implemented in Python
358  (Huerta-Cepas et al., 2016). The circle from inner to outer layer represents classification from kingdom to order
359  successively. Further details on fungal and bacterial taxa at genus level are provided in Fig. S4. The node size represents
360  the average relative abundance of taxa. Only nodes with relative abundance ≥ 1 are highlighted in bold.

## 3.4. Relationship between airborne microbial community abundances and PM$_{10}$ SC species

The NMDS (non-metric multidimensional scaling) ordination exploring the temporal dynamics of microbial community beta diversity among all PM$_{10}$ aerosol samples revealed significant temporal shifts of community structure for both fungi and bacteria (Fig. 4).

An NMDS (two dimensions, stress = 0.15) based on fungal class-level compositions (Fig. 4A) results in three distinct clusters of PM$_{10}$ samples. With one exception (A23), all air samples with higher SC concentration levels (A5 to A20, see Table S2 and Fig. S1) are clustered together and are distinct from those with background levels of atmospheric SC concentrations. This pattern is further confirmed with the analysis of similarity, which shows a significant separation of clusters of samples (ANOSIM; R = 0.31, p < 0.01). As evidenced in Fig. 4A, this difference is mainly explained by the NMDS1 axis, which results from the predominance of only a few class-level fungi in PM$_{10}$ samples, including *Dothideomycetes*, *Tremellomycetes*, *Microbotryomycetes* and *Exobasidiomycetes*. Vector fitting of chemical time series data to the NMDS ordination plot indicates that the latter four fungal community assemblage best correlates with individual SC species. Mannitol (R$^2$ = 0.37, p < 0.01), arabitol (R$^2$ = 0.36, p < 0.01), trehalose (R$^2$ = 0.41, p < 0.01), glucose (R$^2$ = 0.33, p < 0.01), xylitol (R$^2$ = 0.45, p < 0.01), erythritol (R$^2$ = 0.40, p < 0.01) and inositol (R$^2$ = 0.24, p = 0.01) are significantly positively correlated to the fungal assemblage ordination solution.

For bacterial phylum-level compositions (Fig. 4B), an NMDS ordination (two dimensions, stress = 0.07) analysis differentiates the PM$_{10}$ samples into two distinct clusters according to their SC concentrations levels. All air samples with higher SC concentration levels except two (A23 and A24) are clustered separately from those with ambient background concentration levels. ANOSIM analysis (R= 0.69, p < 0.01) further confirms the significant difference between the two clusters of samples. Proteobacteria constitutes the most dominant bacterial phylum during the SC peak over the sampling period. Interestingly, changes in individual SC profiles are significantly correlated with bacterial community temporal shifts (Fig. 4B). Mannitol (R$^2$ = 0.25, p < 0.01), arabitol (R$^2$ = 0.24, p < 0.01), trehalose (R$^2$ = 0.32, p < 0.01), glucose (R$^2$ = 0.32, p < 0.01), xylitol (R$^2$ = 0.38, p < 0.01) and erythritol (R$^2$ = 0.27, p < 0.01) are mainly positively correlated to the bacterial community dissimilarity.

Given the distinct clustering patterns of airborne PM$_{10}$ microbial beta diversity structures according to SC concentration levels, a Pearson's rank correlation analysis has been performed to further examine the relationships between individual SC profiles and airborne microbial community abundance at phylum or class levels. This analysis reveals that for class-level fungi, the abundances of Dothideomycetes, Tremellomycetes and Microbotryomycetes are highly positively correlated (p < 0.05) to the temporal evolutions of the individual SC species concentration levels (Fig. S5A). Likewise, ambient SC species concentration levels are significantly correlated (p < 0.05) to the Proteobacteria phylum (Fig. S5B). To gain further insight into the airborne microbial fingerprints associated with ambient SC species, correlation analyses were also performed at a finer taxonomic level. These analyses show that the temporal dynamics of SC species primarily correlates best (p < 0.05) with the *Cladosporium*, *Alternaria*, *Sporobolomyces* and *Dioszegia* fungal genera (Fig. 5A). Similarly, the time series of SC species are primarily positively correlated (p < 0.05) with *Massilia*, *Pseudomonas*, *Frigoribacterium*, and to a lesser degree (non-significant) with the *Sphingomonas* bacterial genus (Fig. 5B).

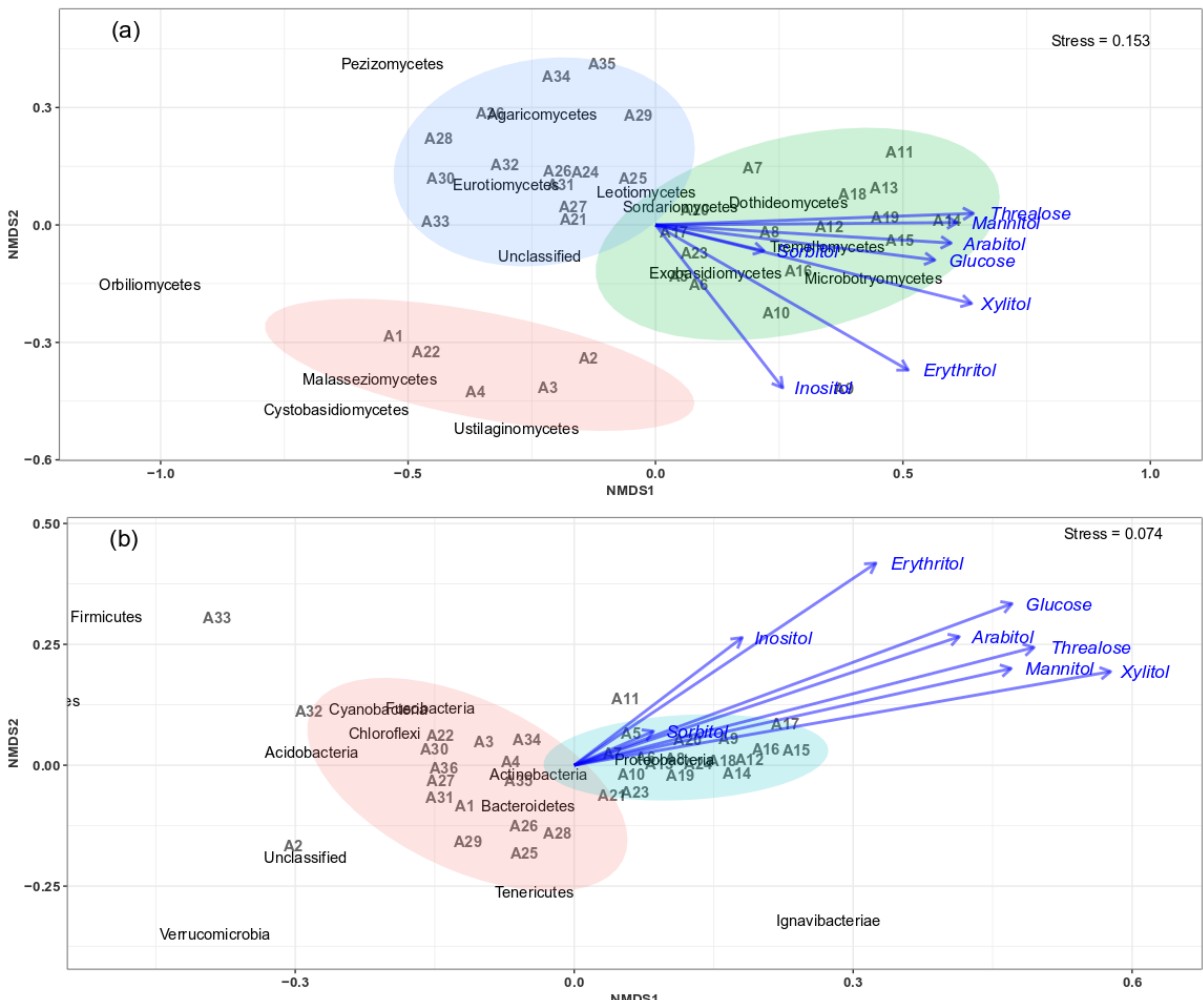

**Figure 4: Main airborne microbial communities associated with atmospheric concentrations of SC species. NMDS ordination plots are used to show relationship among time series of aerosol samples. The stress values indicate an adequate 2-dimensional picture of sample distribution. Ellipses represent 95% confidence intervals for the cluster centroid. NMDS analyses are performed directly on taxonomically assigned quality-filtered sequences tables at class and phylum level respectively for fungi (A) and bacteria (B). Ambient primary sugar concentration levels in PM$_{10}$ appear to be highly influenced by the airborne microbial community structure and abundance. Similar results are obtained with taxonomically assigned MOTU tables, highlighting the robustness of our methodology.**

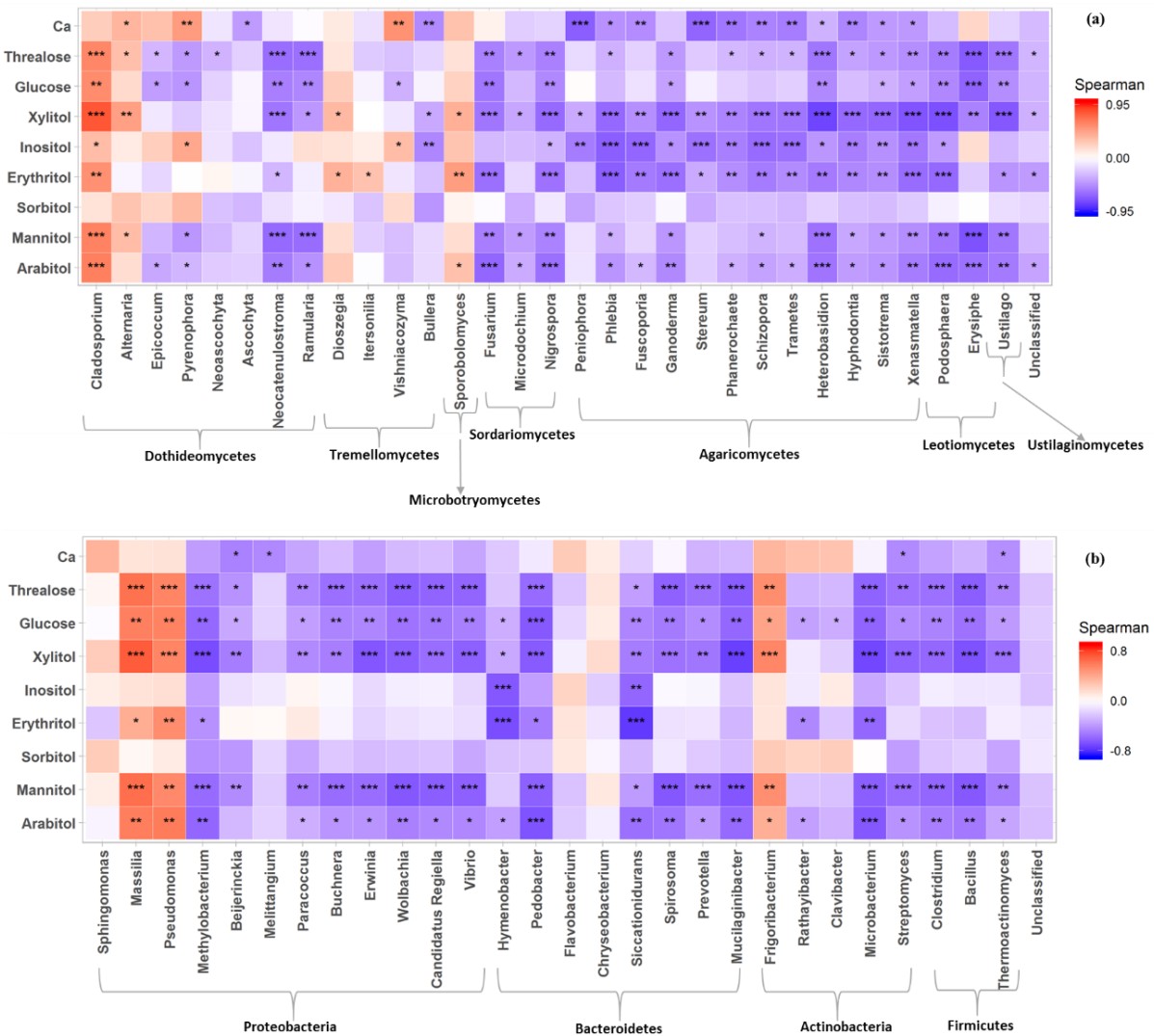

**Figure 5: Heatmap of Spearman's rank correlation between SCs and abundance of airborne communities at the study site. (A) Fungal and (B) bacterial genus, respectively. Only genera with relative abundance ≥ 1 are shown.**

### 3.5. Sources of airborne microbial communities at the study site

As shown in Fig. 6, the airborne microbial genera most positively correlated with SC species are also distributed in the surrounding environmental samples of surface soils and leaves. In addition, microbial taxa of $PM_{10}$ associated with SC species are generally more abundant in the leaves than in the topsoil samples (Fig. 5). In order to further explore and visualize the similarity of species compositions across local environment types, we conducted an NMDS ordination analysis (Fig. 6). As evidenced in Fig. 6, the beta diversities of fungal and bacterial MOTUs are more similar within the same habitat ($PM_{10}$, plant, or soil) and are grouped across habitats as expected. Interestingly, the beta diversities of fungal and bacterial MOTUs in leaf samples and those in airborne $PM_{10}$ are generally not readily distinguishable, with similarity becoming more prominent during atmospheric peaks of SC concentration levels (Fig. 6). However, the overall beta diversities in airborne $PM_{10}$ and in leaf samples are significantly different from those from topsoil samples (ANOSIM, R = 0.89 and 0.80, p < 0.01 for fungal and bacterial communities, respectively), without any overlap regardless of whether or not harvesting activities are performed around the sampling site.

This observation is also confirmed by an unsupervised hierarchical cluster analysis, which reveals a pattern similar to that observed in the NMDS ordination, where taxa from leaf samples and airborne $PM_{10}$ are clustered together, regardless of whether ambient concentration levels of SC peaked or not, and they are clustered separately from those of topsoil samples (Fig. S7).

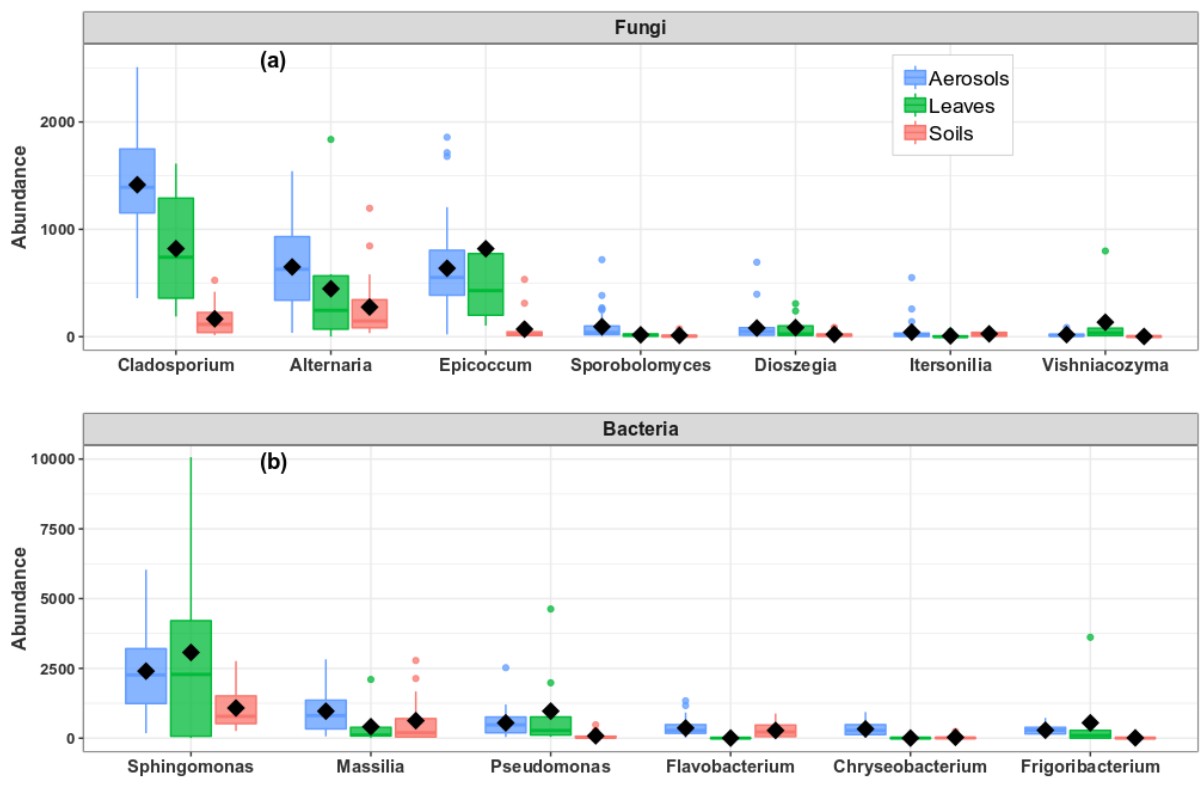


**Figure 6: Abundance of SC species-associated microbial taxa. (A) Fungal and (B) bacterial genera in the airborne PM$_{10}$**
**samples and surrounding environmental samples. Black markers inside each box indicate the mean abundance value,**
**while the top, middle, and bottom lines of the box represent the 75th, median, and 25th percentile, respectively. The**
**whiskers at the top and bottom of the box extend from the 95th to the 5th percent. Data were rarefied at the same**
**minimum sequencing depth.**

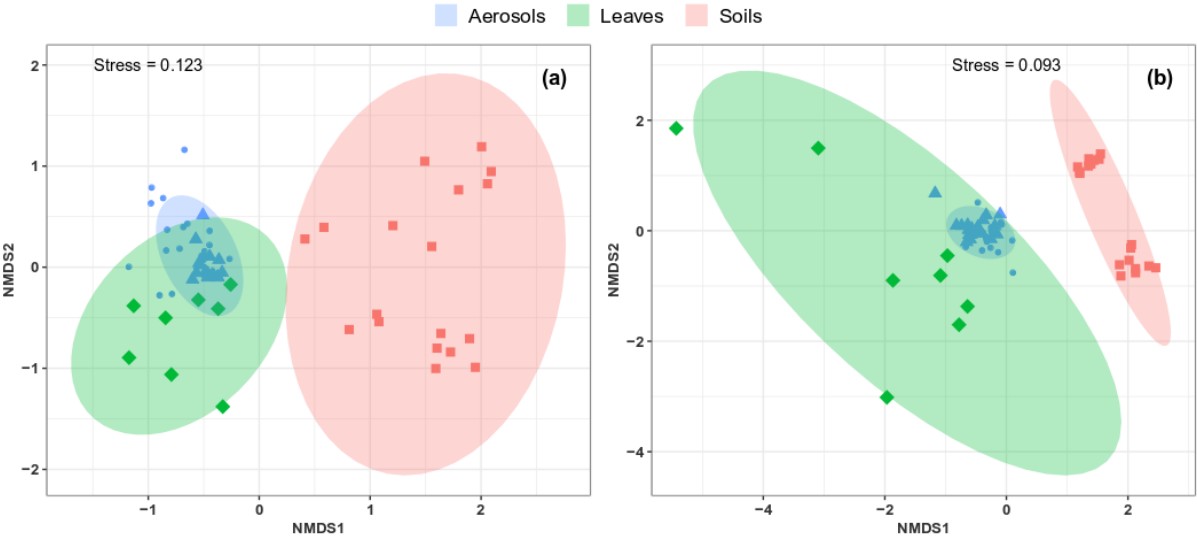


**Figure 7: Compositional comparison of sample types in a NMDS scaling ordination. NDMS plots are constructed from**
**a Horn distance matrix of MOTUs abundances for fungi (A) and bacteria (B), respectively. Data sets are rarefied at the**
**same sequencing depth. The stress values indicate an adequate two-dimensional picture of sample distribution. Ellipses**
**represent 95% confidence intervals for the cluster centroids. Circular and triangular shapes highlight air PM$_{10}$ samples**
**respectively with background and peak SC concentrations.**

## 4. Discussion

Very few studies exist about the interactions between air microbiome and PM chemical profiles (Cao et al., 2014; Elbert et al., 2007). In this study, we used a comprehensive multidisciplinary approach to produce for the first time airborne microbial fingerprints associated with SC species in $PM_{10}$ and to identify the dominant sources of SCs in a continental rural area extensively cultivated.

### 4.1. SCs as a major source of organic matter in $PM_{10}$

SC species have recently been reported to be ubiquitous in $PM_{10}$ collected in several areas in France (Golly et al., 2018; Samaké et al., 2019b). In this study, the total SC presented an average concentration of $259.8 \pm 253.8$ ng m$^{-3}$, with a range of 26.6 to 1,679.5 ng m$^{-3}$ in all air samples. These concentration values are on average five times higher than those typically observed in urban areas in France (average values during summer $48.5 \pm 43.6$ ng.m$^{-3}$) (Golly et al., 2018; Samaké et al., 2019a, 2019b). However, these concentration levels are in agreement with a previous study conducted in a similar environment, i.e., continental rural sites located in large crop fields (Yan et al., 2019).

The total concentrations of SC quantified in the atmospheric $PM_{10}$ over our study site accounted for 0.8 to 13.5% of the daily OM mass. This is remarkable considering that less than 20% of total particulate OM mass can generally be identified at the molecular level. Hence, our results for a nine week-long period indicate that SC could be a major identified molecular fraction of OM for agricultural areas during summer, in agreement with several previous studies conducted worldwide (Jia et al., 2010b; Verma et al., 2018; Yan et al., 2019). Further, it has been shown (Samaké et al., 2019a) that the identified polyols most probably represent only a small fraction of the emission flux from this PBOA source, and that a large fraction of the co-emitted organic material remains unknown. Hence, the PBOA source can potentially represent, for part of the year, a major source of atmospheric OM unaccounted for in CTM models.

### 4.2. Composition of airborne fungal and bacterial communities

In this study, 597 (39-132 MOTUs per sample) and 944 (31-129 MOTUs per sample) MOTUs were obtained for the fungi and bacteria libraries, respectively, reflecting the high richness of airborne microbial communities associated with ambient $PM_{10}$ in a rural agricultural zone in France. Airborne fungi were dominated by Ascomycota (AMC) followed by Basidiomycota (BMC) phyla, consistent with the natural feature of many Ascomycota, whose single-celled or hyphal forms are fairly small to be rapidly aerosolized, in contrast to many Basidiomycota that are typically too large to be easily aerosolized (Moore et al., 2011; Womack et al., 2015). Many members of AMC and BMC are well known to actively eject ascospores and basidiospores as well as aqueous jets and droplets containing a mixture of carbohydrates and inorganic solutes into the atmosphere (Elbert et al., 2007; Womack et al., 2015). The prevalence of Ascomycota and Basidiomycota is consistent with results from previous studies also indicating that the Dikarya subkingdom (Ascomycota and Basidiomycota) represents about 98% of known species in the biological Kingdom of Eumycota (i.e., fungi) in atmosphere (Elbert et al., 2007; James et al., 2006; Womack et al., 2015; Xu et al., 2017)

Airborne bacteria in this study belonged mainly to the Proteobacteria, Bacteroidetes, Actinobacteria and Firmicutes phyla, consistent with previous studies (Liu et al., 2019; Maron et al., 2005; Wei et al., 2019b). Gram-negative Proteobacteria constitute a major taxonomic group among prokaryotes (Itävaara et al., 2016; Yadav et al., 2018), which includes bacterial taxa very diverse and important in agriculture, capable of fixing nitrogen in symbiosis with plants (Itävaara et al., 2016; Yadav et al., 2018). Proteobacteria can survive under conditions with very low nutrient content, which explains their atmospheric versatility (Itävaara et al., 2016; Yadav et al., 2018). These results are similar to those observed in previous studies conducted in different environments around the world, where Proteobacteria, Actinobacteria and Firmicutes have also been reported as dominant bacterial phyla (Liu et al., 2019; Maron et al., 2005; Wei et al., 2019a). In particular, the most frequent gram-negative (Proteobacteria and Bacteroidetes) and gram-positive (Actinobacteria and Firmicutes) bacteria, and filamentous fungi (Ascomycota and Basidiomycota) have been previously linked to raw straw handling activities. For instance, it has been suggested that straw combustion during agricultural activities could be a major source of airborne microorganisms in $PM_{2.5}$ at the northern plains of China (Wei et al., 2019a, 2019b). However, in our study, SC species are not correlated (R = −0.09, p = 0.46; Fig. S7) with levoglucosan during the campaign period, confirming that biomass burning is not an important source of airborne microbial taxa associated with SCs in our $PM_{10}$ series.

Bubble bursting associated with sea spray could also potentially be a source of bacteria, fungi and water-soluble
organic species, along with sea salts, to $PM_{10}$ (Prather et al., 2013; Zhu et al., 2015). However, SC species were
not found to be significantly related to $Cl^-$ (R = −0.14, p = 0.28) or $Na^+$ (R = −0.18, p = 0.16), which are two
inorganic tracers typical of marine sources; nor correlated with methanesulfonic acid (R = −0.05, p = 0.69), a well-
known tracer of biogenic marine activity (Arndt et al., 2017; Gaston et al., 2010). It therefore seems unlikely that
the sources of SCs from marine environments were significant at this site. This point is further discussed in Sect.
498  4.4.

**4.3. Atmospheric concentration levels of SC species in $PM_{10}$ are associated with the abundance of few specific airborne taxa of fungi and bacteria**

SCs are widely produced in large quantities by many microorganisms to cope with environmental stress conditions
(Medeiros et al., 2006). SC species are known to accumulate at high concentrations in microorganisms at low
water availability to reduce intracellular water activity and prevent enzyme inhibition due to dehydration
(Hrynkiewicz et al., 2010). In addition, temporal dynamics of ambient polyols concentrations have been suggested
as an indicator to follow the general seasonal trend in airborne fungal spore counts (Bauer et al., 2008; Gosselin et
al., 2016). Although this strategy has allowed introducing conversion ratios between specific polyols species (i.e.,
arabitol and mannitol) and airborne fungal spores in general (Bauer et al., 2008), the structure of the airborne
microbial community associated with SC species has not yet been studied. Our results provide culture-independent
evidence that the airborne microbiome structure and the combined bacterial and fungal communities largely
determine the SC species concentration levels in $PM_{10}$.
Temporal fluctuations in the abundance of only few specific fungal and bacterial genera reflect the temporal
dynamics of ambient SC concentrations. For fungi, genera that show a significant positive correlation (p < 0.05)
with SC species includes *Cladosporium*, *Alternaria, Sporobolomyces* and *Dioszegia. Cladosporium* and
*Alternaria,* are fungal genera that contribute on average to 47.9% of total fungal sequence reads in our air samples
series. These are asexual fungal genera that produce spores by dry-discharge mechanisms wherein spores are
detached from their parent colonies and easily dispersed by the ambient air flow or other external forces (e.g.,
raindrops, elevated temperature, etc.), as opposed to actively discharged spores with liquid jets or droplets into the
air (Elbert et al., 2007; Wei et al., 2019b; Womack et al., 2015). Our results are consistent with the well-known
seasonal behavior of airborne fungal spores, with levels of *Cladosporium* and *Alternaria* which have been shown
to reach their maximum from early to midsummer in a rural agricultural area of Portugal (Oliveira et al., 2009).
Similarly, bacterial genera positively correlated with SC species are *Massilia*, *Pseudomonas*, *Frigoribacterium*,
and *Sphingomonas*. Although it is the prevalent bacterial genus at the study site, *Sphingomonas* is indeed not
significantly positively correlated with SC species. The genus *Sphingomonas* is well-known to include numerous
metabolically versatile species capable of using carbon compounds usually present in the atmosphere (Cáliz et al.,
2018). The atmospheric abundance of species affiliated with *Massilia* has already been linked to the change in the
stage of plant development (Ofek et al., 2012), which can be attributed to the capacity of *Massilia* to promote plant
growth, through the production of indole acetic acid (Kuffner et al., 2010), or siderophores (Hrynkiewizc et al.,
2010), and to be antagonist towards *Phytophthora infestans* (Weinert et al., 2010).
As far as we know, this is the first study evaluating microbial fingerprints with SC species in atmospheric PM,
hence it is not possible to compare our correlation results with that of previous works. However, it has already
been suggested that types and quantities of SC species produced by fungi under culture conditions are specific to
microbial species and external conditions such as carbon source, drought and heat, etc. (Hrynkiewicz et al., 2010).
In future studies, we intend to apply a culture-dependent method to directly characterize the SC contents of some
species amongst the dominant microbial taxa identified in this study after growth under several laboratory
chambers reproducing controlled environmental conditions in terms of temperature, water vapor or carbon sources.

**4.4. Local vegetation as major source of airborne microbial taxa of $PM_{10}$ associated with SC species**

There are still many challenging questions on the emission processes leading to fungi and bacteria being introduced
into the atmosphere, together with their chemical components. In particular, the potential influence of soil and
vegetation and their respective roles in structuring airborne microbial communities is still debated
(Lymperopoulou et al., 2016; Rathnayake et al., 2016; Womack et al., 2015), especially since this knowledge is
particularly essential for the precise modeling of PBOA emissions processes to the atmosphere within Chemical
Transport Models.
Characterization of the temporal dynamics of SC species concentrations could provide important information on
PBOA sources in terms of composition, environmental drivers and impacts. The results obtained over a nine week-
period of daily $PM_{10}$ SC measurements clearly show that the temporal dynamics of sorbitol (R= 0.59, p < 0.001)
and inositol (R= 0.64, p < 0.001) are well correlated linearly with that of calcium, a typical inorganic water-soluble
ion from crustal material. This indicates a common atmospheric origin for these chemicals. Sorbitol and inositol
are well-known reduced sugars that serve as carbon source for microorganisms when other carbon sources are
limited (Ng et al., 2018; Xue et al., 2010). In microorganisms, sorbitol and inositol are mainly produced by the
reduction of intracellular glucose by aldose reductase in the cytoplasm (Ng et al., 2018; Welsh, 2000; Xue et al.,
2010). Moreover, significant concentrations of both sorbitol and inositol have already been measured in surface
soil samples from five cultivated fields in the San Joaquin Valley, USA (Jia et al., 2010b; Medeiros et al., 2006).
Therefore, sorbitol and inositol are most likely associated with microorganisms from soil resuspension.
With the exception of sorbitol and inositol, all other SC species measured in air samples at our sampling site are
strongly correlated with each other, indicating a common origin. Daily calcium concentration peaks are not
systematically associated with those of these other SC species. Interestingly, the highest atmospheric levels of
these SC species occurred on August 8$^{th}$ 2017, coinciding well with daily harvesting activities around the site. This
is also consistent with a multi-year monitoring of the dominant SCs in $PM_{10}$ at this site, where ambient SCs showed
a clear seasonal trend with higher values recorded in early August and in good agreement with harvesting activities
around the study area every year from 2012 to 2017 (Samaké et al., 2019a). This suggests that the processes
responsible for the dynamics of atmospheric concentrations of SCs are replicated annually and most likely
effective over large areas of field crop (Golly et al., 2018; Samaké et al., 2019a). Interestingly, glucose—the most
common monosaccharide present in vascular plants and microorganisms— has already been proposed as
molecular indicator of biota emitted into the atmosphere by vascular plants and/or by the resuspension of soil from
agricultural land (Jia et al., 2010b; Pietrogrande et al., 2014). Therefore, all other SC species measured in our series
can be considered to be most likely the result of the mechanical resuspension of crop residues (e.g., leaf debris)
and microorganisms attached to them. Other confirmations of this interpretation stem from the excellent daily co-
variations observed in the $PM_{10}$ between SC species levels and ambient cellulose, widely considered as a reliable
indicator of the plant debris source in PM studies (Bozzetti et al., 2016; Hiranuma et al., 2019).
Microbial abundance and community structure in samples from the surrounding environment can provide further
useful information on sources apportionment and importance. Our data indicates that the airborne microbial genera
most positively correlated to SC species are also distributed in surrounding environmental samples from both
surface soils and leaves, suggesting a dominant influence of the local environments for microbial taxa associated
with SC species, as opposed to long-range transport. This observation makes sense since actively discharged
ascospores and basidiospores are generally relatively large airborne particles with short atmospheric residence
time (Elbert et al., 2007; Womack et al., 2015), limiting the possibilities of long-range dissemination. Accordingly,
the majority of previous studies investigating the potential sources of air microbes identified the local surface
environments (e.g., leaves, soils, etc.) to have more important effects on airborne microbiome structure in field
crop areas (Bowers et al., 2011; Wei et al., 2019b; Womack et al., 2015). This is all the more the case in our study,
with homogeneous crop activities for 10's to 100's of km around the site.
In the present study, microbial diversity and richness observed in the surface soils are generally higher than those
in leaf surfaces. Microbial taxa most positively correlated with $PM_{10}$ SC species are generally more abundant in
leaf than in topsoil samples. These results were unexpected and show the possible importance of leaf surfaces in
structuring the airborne taxa associated with SC species. Considering the general grouping of leaf samples and
airborne $PM_{10}$ regardless of harvesting activities around the study site in addition to the separate assemblies of
rarefied MOTUs in airborne $PM_{10}$ and topsoil samples, it can be argued that aerial parts of plants are the major
source of microbial taxa associated with SC species. Such observation is most likely related to increased vegetative
surface (e.g., leaves) in summer that provides sufficient nutrient resources for microbial growth (Rathnayake et
al., 2016). By reviewing previous studies, *Alternaria* and *Epicocum*, which made 30% of total fungal sequence
reads in all air samples in this study, have been shown to be common saprobes or weak pathogens of leaf surfaces
(Andersen et al., 2009). Similarly, *Cladosporium*, which accounted for 32.9% of total fungal genera in all air
samples, have also been shown to be a common saprotrophic fungi inhabiting in decayed tree or plant debris (Wei
et al., 2019b). The high relative abundance of *Sphingomonas* and *Massilia*, accounting for 28.4% of total bacterial
genera in all air samples, is also noticeable. These two phyllosphere inhabiting bacterial genera are well-known
for their plant protective potential against phytopathogens (Aydogan et al., 2018; Rastogi et al., 2013).
Altogether, these observations support our interpretation that leaves are the major direct source of airborne fungi
and bacteria during the summer months at this site of large agricultural activities. Endophytes and epiphytes can
be dispersed in the air and transported vertically as particles by the air currents, much faster and more widely than
by other mechanisms, such as direct dissemination from surface soil, which is generally controlled by soil moisture
(Jocteur Monrozier et al., 1993). The most wind-dispersible soil constituents are indeed the smallest soil particles
(i.e. clay-size fraction), which contain the largest number of microorganisms (Jocteur Monrozier et al., 1993) and
can only be released into the atmosphere under conditions of prolonged drought. This interpretation is also
consistent with previous studies (Bowers et al., 2011; Liu et al., 2019; Lymperopoulou et al., 2016; Mhuireach et
al., 2016), which also show the extent to which endophytes and epiphytes can serve as quantitatively important
sources of airborne microbes during summertime when vegetation density is highest. For example,
Lymperopoulou et al. (2016) observed that bacteria and fungi suspended in the air are generally two to more than
ten times more abundant in air that passed over 50 m of vegetated surface than that is immediately upwind of the
same vegetated surface. However, the relatively abundance of taxa associated with SCs in surface soils in this
study could also be indicative of a feedback loop in which the soil may serve as sources of microbial endophytes
and epiphytes for plants while the local vegetation in turns may serve as sources and sinks of microbes for local
soils during leaf senescence.

## 5. Conclusion

Primary biogenic organic aerosols (PBOA) affects human health, climate, agriculture, etc. However, the details of
microbial communities associated with the temporal and spatial variations in atmospheric concentrations of SC,
tracers of PBOA, remain unknown. The present study aimed at identifying the airborne fungi and bacteria
associated with SC species in $PM_{10}$ and their major sources in the surrounding environment (soils and vegetation).
To that end, we combined high-throughput sequencing of bacteria and fungi with detailed physicochemical
characterization of $PM_{10}$ soils and leaf samples collected at a continental rural background site located in a large
agricultural area in France.
The main results demonstrate that the identified SC species are a major contributor of OM in summer, accounting
together for 0.8 to 13.5% of OM mass in air. The atmospheric concentration peaks of SC are coincident the daily
harvest activities around the sampling site, pointing towards direct resuspension of biological materials, i.e. crop
residues and associated microbiota as an important source of SC in our $PM_{10}$ series. Furthermore, we have also
discovered that the temporal evolutions of SC in $PM_{10}$ are associated with the abundance of only few specific
airborne fungi and bacteria taxa. These microbial taxa are significantly enhanced in the surrounding environmental
samples of leaves over surface soils. Finally, the excellent correlation of SC species and cellulose, a marker of
plant materials, implies that local vegetation is likely the most important source of fungi and bacteria taxa
associated with SC in $PM_{10}$ at rural locations directly influenced by agricultural activities in France.
Our findings is a first step in the understanding of the processes leading to the emission of these important chemical
species and large OM fraction of PM in the atmosphere, and to the parametrization of these processes for their
introduction in CTM models. They could also be used for planning efforts to reduce both the PBOA source
strengths and the spreading of airborne microbial and derivative allergens such as endotoxins, mycotoxins, etc.
However, it remains to investigate how-well different climate patterns and sampling site specificities, in terms of
land use and vegetation cover, could affect our main conclusions.

**Data and materials availability:** The sequencing data files are available from the DRYAD repository
(doi:10.5061/dryad.2fqz612m4). All relevant chemical and environmental data sets are archived at the IGE
(Institut des Géosciences de l'Environnement), and are available upon request from the co-author (Jean-Luc
Jaffrezo).
**Competing interests:** The authors declare that they have no competing interests.
**Author contributions:** J.-L.J., J-M-F.M., G.U. supervised the thesis of A.S. and J.-L.J., J.M.F.-M., G.U and A.S.
designed the research project. P.T. gives advice for soils and leaves sampling. S.C. supervised the sample
collections and provided the agricultural activity records. V.J. developed the analytical techniques for SC species
and cellulose measurements. A.S. and A.B. performed the experiments. A.B. performed the bioinformatic
analyses. A.S. performed statistical analyses and wrote the original manuscript draft. S.W. produced the circular
phylogenetic trees. All authors reviewed and edited the final manuscript.
**Acknowledgements:** We acknowledge the work of many engineers in the lab at the Institut des Géosciences de
l'Environnement for the analyses (Anthony Vella, Vincent Lucaire). The authors would like to kindly thank the
dedicated efforts of many other people at the sampling site and in the laboratories for collecting and analyzing the
samples.
**Financial support:** The PhD of Abdoulaye Samaké is funded by the Government of Mali. We gratefully
acknowledge the LEFE-CHAT and EC2CO programs of the CNRS for financial supports of the CAREMBIOS
multidisciplinary project, with ADEME funding. Chemical and microbiological analytical aspects were supported
at IGE by the Air-O-Sol and MOME platforms, respectively, within Labex OSUG@2020 (ANR10 LABX56).

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

Methods to investigate the global atmospheric microbiome, Front. Microbiol., 10, 243,
doi:10.3389/fmicb.2019.00243, 2019.
Elbert, W., Taylor, P. E., Andreae, M. O., and Pöschl, U.: Contribution of fungi to primary biogenic aerosols in
the atmosphere: wet and dry discharged spores, carbohydrates, and inorganic ions, Atmos. Chem. Phys., 7(17),
4569–4588, doi:10.5194/acp-7-4569-2007, 2007.
Ficetola, G., Coissac, E., Zundel, S., Riaz, T., Shehzad, W., Bessière, J., Taberlet, P., and Pompanon, F.: An in
silico approach for the evaluation of DNA barcodes, BMC Genomics, 11(1), 434, doi:10.1186/1471-2164-11-434,
718    2010.

Fortenberry, C. F., Walker, M. J., Zhang, Y., Mitroo, D., Brune, W. H., and Williams, B. J.: Bulk and molecular-
level characterization of laboratory-aged biomass burning organic aerosol from oak leaf and heartwood fuels,
Atmos. Chem. Phys., 18(3), 2199–2224, doi:10.5194/acp-18-2199-2018 2018.
Fröhlich-Nowoisky, J., Kampf, C. J., Weber, B., Huffman, J. A., Pöhlker, C., Andreae, M. O., Lang-Yona, N.,
Burrows, S. M., Gunthe, S. S., Elbert, W., Su, H., Hoor, P., Thines, E., Hoffmann, T., Després, V. R., and Pöschl,
U.: Bioaerosols in the earth system: climate, health, and ecosystem interactions, Atmos. Res., 182, 346–376,
doi:10.1016/j.atmosres.2016.07.018, 2016.
Fu, P., Kawamura, K., Kobayashi, M., and Simoneit, B. R.: Seasonal variations of sugars in atmospheric particulate
matter from Gosan, Jeju Island: Significant contributions of airborne pollen and Asian dust in spring, Atmos.
environ., 55, 234–239, doi:10.1029/2003JD003697, 2012.
Fuzzi, S., Andreae, M. O., Huebert, B. J., Kulmala, M., Bond, T. C., Boy, M., Doherty, S. J., Guenther, A.,
Kanakidou, M., and Kawamura, K.: Critical assessment of the current state of scientific knowledge, terminology,
and research needs concerning the role of organic aerosols in the atmosphere, climate, and global change, Atmos.
Chem. Phys, 22, doi:10.5194/acp-6-2017-2006, 2006.
Fuzzi, S., Baltensperger, U., Carslaw, K., Decesari, S., Denier van der Gon, H., Facchini, M. C., Fowler, D., Koren,
I., Langford, B., Lohmann, U., Nemitz, E., Pandis, S., Riipinen, I., Rudich, Y., Schaap, M., Slowik, J. G.,
Spracklen, D. V., Vignati, E., Wild, M., Williams, M., and Gilardoni, S.: Particulate matter, air quality and climate:
lessons learned and future needs, Atmos. Chem. Phys., 15(14), 8217–8299, doi:10.5194/acp-15-8217-2015, 2015.
Gaston, C. J., Pratt, K. A., Qin, X., and Prather, K. A.: Real-time detection and mixing state of methanesulfonate
in single particles at an inland urban location during a phytoplankton bloom, Environ. Sci. Technol., 44(5), 1566–
1572, doi:10.1021/es902069d, 2010.
Golly, B., Waked, A., Weber, S., Samaké, A., Jacob, V., Conil, S., Rangognio, J., Chrétien, E., Vagnot, M.-P.,
Robic, P.-Y., Besombes, J.-L., and Jaffrezo, J.-L.: Organic markers and OC source apportionment for seasonal
variations of $PM_{2.5}$ at 5 rural sites in France, Atmos. Environ., 198, 142–157, doi:10.1016/j.atmosenv.2018.10.027,
743    2018.

Gosselin, M. I., Rathnayake, C. M., Crawford, I., Pöhlker, C., Fröhlich-Nowoisky, J., Schmer, B., Després, V. R.,
Engling, G., Gallagher, M., Stone, E., Pöschl, U., and Huffman, J. A.: Fluorescent bioaerosol particle, molecular
tracer, and fungal spore concentrations during dry and rainy periods in a semi-arid forest, Atmos. Chem. Phys.,
16(23), 15165–15184, doi:10.5194/acp-16-15165-2016, 2016.
Heald, C. L., Coe, H., Jimenez, J. L., Weber, R. J., Bahreini, R., Middlebrook, A. M., Russell, L. M., Jolleys, M.,
Fu, T.-M., Allan, J. D., Bower, K. N., Capes, G., Crosier, J., Morgan, W. T., Robinson, N. H., Williams, P. I.,
Cubison, M. J., DeCarlo, P. F., and Dunlea, E. J.: Exploring the vertical profile of atmospheric organic aerosol:
comparing 17 aircraft field campaigns with a global model, Atmos. Chem. Phys., 11(24), 12673–12696,
doi:10.5194/acp-11-12673-2011, 2011.
Hill, T. C. J., DeMott, P. J., Conen, F., and Möhler, O.: Impacts of bioaerosols on atmospheric ice nucleation
Processes, in Microbiology of Aerosols, pp. 195–219, John Wiley & Sons, Ltd.,
https://doi.org/10.1002/9781119132318, 2017.
Hiranuma, N., Adachi, K., Bell, D. M., Belosi, F., Beydoun, H., Bhaduri, B., Bingemer, H., Budke, C., Clemen,
H.-C., Conen, F., Cory, K. M., Curtius, J., DeMott, P. J., Eppers, O., Grawe, S., Hartmann, S., Hoffmann, N.,
Höhler, K., Jantsch, E., Kiselev, A., Koop, T., Kulkarni, G., Mayer, A., Murakami, M., Murray, B. J., Nicosia, A.,
Petters, M. D., Piazza, M., Polen, M., Reicher, N., Rudich, Y., Saito, A., Santachiara, G., Schiebel, T., Schill, G.
P., Schneider, J., Segev, L., Stopelli, E., Sullivan, R. C., Suski, K., Szakáll, M., Tajiri, T., Taylor, H., Tobo, Y.,
Ullrich, R., Weber, D., Wex, H., Whale, T. F., Whiteside, C. L., Yamashita, K., Zelenyuk, A., and Möhler, O.: A
comprehensive characterization of ice nucleation by three different types of cellulose particles immersed in water,
Atmos. Chem. Phys., 19(7), 4823–4849, doi:10.5194/acp-19-4823-2019, 2019.
Hrynkiewicz, K., Baum, C., and Leinweber, P.: Density, metabolic activity, and identity of cultivable rhizosphere
bacteria on Salix viminalis in disturbed arable and landfill soils, J. Plant Nutr. Soil Sci., 173(5), 747–756,
doi:10.1002/jpln.200900286, 2010.
Hsieh, T. C., Ma, K. H. and Chao, A.: iNEXT: an R package for rarefaction and extrapolation of species diversity
(Hill numbers), Methods Ecol. Evol., 7(12), 1451–1456, doi:10.1111/2041-210X.12613, 2016.
Huerta-Cepas, J., Serra, F. and Bork, P.: ETE 3: Reconstruction, analysis, and visualization of Phylogenomic Data,
Mol. Biol. Evol., 33(6), 1635–1638, doi:10.1093/molbev/msw046, 2016.
Huffman, J. A. and Santarpia, J.: Online techniques for quantification and characterization of biological aerosols,
in Microbiology of Aerosols, pp. 83–114, John Wiley & Sons, Ltd., https://doi.org/10.1002/9781119132318, 2017.
Huffman, J. A., Perring, A. E., Savage, N. J., Clot, B., Crouzy, B., Tummon, F., Shoshanim, O., Damit, B.,
Schneider, J., Sivaprakasam, V., Zawadowicz, M. A., Crawford, I., Gallagher, M., Topping, D., Doughty, D. C.,
Hill, S. C., and Pan, Y.: Real-time sensing of bioaerosols: review and current perspectives, Aerosol Science and
Technology, 1–31, doi:10.1080/02786826.2019.1664724, 2019.
Itävaara, M., Salavirta, H., Marjamaa, K., and Ruskeeniemi, T.: Geomicrobiology and metagenomics of Terrestrial
Deep Subsurface Microbiomes, in Advances in Applied Microbiology, vol. 94, pp. 1–77, Elsevier.,
doi:10.1016/bs.aambs.2015.12.001, 2016.
James, T. Y., Kauff, F., Schoch, C. L., Matheny, P. B., Hofstetter, V., Cox, C. J., Celio, G., Gueidan, C., Fraker,
E., Miadlikowska, J., Lumbsch, H. T., Rauhut, A., Reeb, V., Arnold, A. E., Amtoft, A., Stajich, J. E., Hosaka, K.,
Sung, G.-H., Johnson, D., O'Rourke, B., Crockett, M., Binder, M., Curtis, J. M., Slot, J. C., Wang, Z., Wilson, A.
W., Schüßler, A., Longcore, J. E., O'Donnell, K., Mozley-Standridge, S., Porter, D., Letcher, P. M., Powell, M.
J., Taylor, J. W., White, M. M., Griffith, G. W., Davies, D. R., Humber, R. A., Morton, J. B., Sugiyama, J.,
Rossman, A. Y., Rogers, J. D., Pfister, D. H., Hewitt, D., Hansen, K., Hambleton, S., Shoemaker, R. A.,
Kohlmeyer, J., Volkmann-Kohlmeyer, B., Spotts, R. A., Serdani, M., Crous, P. W., Hughes, K. W., Matsuura, K.,
Langer, E., Langer, G., Untereiner, W. A., Lücking, R., Büdel, B., Geiser, D. M., Aptroot, A., Diederich, P.,
Schmitt, I., Schultz, M., Yahr, R., Hibbett, D. S., Lutzoni, F., McLaughlin, D. J., Spatafora, J. W., and Vilgalys,
R.: Reconstructing the early evolution of Fungi using a six-gene phylogeny, Nature, 443(7113), 818–822,
doi:10.1038/nature05110, 2006.
Jia, Y., Bhat, S., and Fraser, M. P.: Characterization of saccharides and other organic compounds in fine particles
and the use of saccharides to track primary biologically derived carbon sources, Atmos. Environ., 44(5), 724–732,
doi:10.1016/j.atmosenv.2009.10.034, 2010a.
Jia, Y., Clements, A. L. and Fraser, M. P.: Saccharide composition in atmospheric particulate matter in the
southwest US and estimates of source contributions, J. Aerosol Sci., 41(1), 62–73,
doi:10.1016/j.jaerosci.2009.08.005, 2010b.
Jiang, W., Liang, P., Wang, B., Fang, J., Lang, J., Tian, G., Jiang, J., and Zhu, T. F.: Optimized DNA extraction
and metagenomic sequencing of airborne microbial communities, Nat. Protoc., 10(5), 768–779,
doi:10.1038/nprot.2015.046, 2015.
Jocteur Monrozier, L., Guez, P., Chalamet, A., Bardin, R., Martins, J., and Gaudet, J. P.: Distribution of
microorganisms and fate of xenobiotic molecules in unsaturated soil environments, Sci. Total Environ., 136(1–2),
121–133, doi:10.1016/0048-9697(93)90302-M, 1993.
Kang, M., Ren, L., Ren, H., Zhao, Y., Kawamura, K., Zhang, H., Wei, L., Sun, Y., Wang, Z., and Fu, P.: Primary
biogenic and anthropogenic sources of organic aerosols in Beijing, China: insights from saccharides and n-alkanes,
Environ. Pollut., 243, 1579–1587, doi:10.1016/j.envpol.2018.09.118, 2018.
Kelly, F. J. and Fussell, J. C.: Air pollution and public health: emerging hazards and improved understanding of
risk, Environ. Geochem. Health, 37(4), 631–649, doi:10.1007/s10653-015-9720-1, 2015.
Kuffner, M., De Maria, S., Puschenreiter, M., Fallmann, K., Wieshammer, G., Gorfer, M., Strauss, J., Rivelli, A.
R., and Sessitsch, A.: Culturable bacteria from Zn- and Cd-accumulating Salix caprea with differential effects on
plant growth and heavy metal availability, J. Appl. Microbiol., 108(4), 1471–1484, doi:10.1111/j.1365-
2672.2010.04670.x, 2010.
Lecours, P. B., Duchaine, C., Thibaudon, M., and Marsolais, D.: Health impacts of bioaerosol exposure, in
Microbiology of Aerosols, pp. 249–268, John Wiley & Sons, Ltd., https://doi.org/10.1002/9781119132318, 2017.
Liu, H., Hu, Z., Zhou, M., Hu, J., Yao, X., Zhang, H., Li, Z., Lou, L., Xi, C., Qian, H., Li, C., Xu, X., Zheng, P.,
and Hu, B.: The distribution variance of airborne microorganisms in urban and rural environments, Environ.
Pollut., 247, 898–906, doi:10.1016/j.envpol.2019.01.090, 2019.
Luhung, I., Wu, Y., Ng, C. K., Miller, D., Cao, B., and Chang, V. W.-C.: Protocol improvements for low
concentration DNA-based bioaerosol sampling and analysis, PLOS ONE, 10(11), e0141158,
doi:10.1371/journal.pone.0141158, 2015.
Lymperopoulou, D. S., Adams, R. I., and Lindow, S. E.: Contribution of vegetation to the microbial composition
of nearby outdoor air, Appl. Environ. Microbiol., 82(13), 3822–3833, doi:10.1128/AEM.00610-16, 2016.
Maron, P.-A., Lejon, D. P. H., Carvalho, E., Bizet, K., Lemanceau, P., Ranjard, L., and Mougel, C.: Assessing
genetic structure and diversity of airborne bacterial communities by DNA fingerprinting and 16S rDNA clone
library, Atmos. Environ., 39(20), 3687–3695, doi:10.1016/j.atmosenv.2005.03.002, 2005.
McMurdie, P. J. and Holmes, S.: phyloseq: An R package for reproducible interactive analysis and graphics of
microbiome sensus Data, PLoS ONE, 8(4), e61217, doi:10.1371/journal.pone.0061217, 2013.
Medeiros, P. M., Conte, M. H., Weber, J. C., and Simoneit, B. R. T.: Sugars as source indicators of biogenic
organic carbon in aerosols collected above the howland experimental forest, Maine, Atmos. Environ., 40(9), 1694–
1705, doi: doi.org/10.1016/j.atmosenv.2005.11.001, 2006.
Mercier, C., Boyer, F., Kopylova, E., Taberlet, P., Bonin, A., and Coissac, E.: SUMATRA and SUMACLUST:
fast and exact comparison and clustering of sequences. Programs and Abstracts of the SeqBio, Workshop, pp. 27–
832 29., 2013.

Mhuireach, G., Johnson, B. R., Altrichter, A. E., Ladau, J., Meadow, J. F., Pollard, K. S., and Green, J. L.: Urban
greenness influences airborne bacterial community composition, Sci. Total Environ., 571, 680–687,
doi:10.1016/j.scitotenv.2016.07.037, 2016.
Moore, D., Robson, G. D., and Trinci, A. P. J.: 21st century guidebook to fungi, Cambridge University Press,
Cambridge., https://doi.org/10.1017/CBO9780511977022, 2011.
Ng, T. W., Ip, M., Chao, C. Y. H., Tang, J. W., Lai, K. P., Fu, S. C., Leung, W. T., and Lai, K. M.: Differential
gene expression in Escherichia coli during aerosolization from liquid suspension, Appl. Microbiol. Biotechnol.,
102(14), 6257–6267, doi:10.1007/s00253-018-9083-5, 2018.
Ofek, M., Hadar, Y. and Minz, D.: Ecology of root colonizing massilia (oxalobacteraceae), PLoS ONE, 7(7),
e40117, doi:10.1371/journal.pone.0040117, 2012.
Oksanen, J., Blanchet, F. G., Friendly, M., Kindt, R., Legendre, P., McGlinn, D., R. Minchin, P. R., O'Hara, R.
B., Simpson, G. L., Solymos, Peter, Stevens, M. H. H., Szoecs, E., and Wagner, H.: Vegan: community ecology
package, https://cran.r-project.org, https://github.com/vegandevs/vegan, 2019.
Oliveira, M., Ribeiro, H., Delgado, J. L., and Abreu, I.: The effects of meteorological factors on airborne fungal
spore concentration in two areas differing in urbanisation level, Int. J. Biometeorol., 53(1), 61–73,
doi:10.1007/s00484-008-0191-2, 2009.
Prather, K. A., Bertram, T. H., Grassian, V. H., Deane, G. B., Stokes, M. D., DeMott, P. J., Aluwihare, L. I.,
Palenik, B. P., Azam, F., Seinfeld, J. H., Moffet, R. C., Molina, M. J., Cappa, C. D., Geiger, F. M., Roberts, G. C.,
Russell, L. M., Ault, A. P., Baltrusaitis, J., Collins, D. B., Corrigan, C. E., Cuadra-Rodriguez, L. A., Ebben, C. J.,
Forestieri, S. D., Guasco, T. L., Hersey, S. P., Kim, M. J., Lambert, W. F., Modini, R. L., Mui, W., Pedler, B. E.,
Ruppel, M. J., Ryder, O. S., Schoepp, N. G., Sullivan, R. C., and Zhao, D.: Bringing the ocean into the laboratory
to probe the chemical complexity of sea spray aerosol, Proc Natl Acad Sci U S A, 110(19), 7550–7555,
doi:10.1073/pnas.1300262110, 2013.
Pietrogrande, M. C., Bacco, D., Visentin, M., Ferrari, S., and Casali, P.: Polar organic marker compounds in
atmospheric aerosol in the Po valley during the Supersito campaigns — part 2: seasonal variations of sugars,
Atmos. Environ., 97, 215–225, doi:0.1016/j.atmosenv.2014.07.056, 2014.

Pope, C. A. and Dockery, D. W.: Health effects of fine particulate air pollution: lines that connect, J. Air Waste Manag. Assoc., 56(6), 709–742, 2006.

Rastogi, G., Coaker, G. L., and Leveau, J. H. J.: New insights into the structure and function of phyllosphere microbiota through high-throughput molecular approaches, FEMS Microbiol. Lett., 348(1), 1–10, doi:10.1111/1574-6968.12225, 2013.

Rathnayake, C. M., Metwali, N., Baker, Z., Jayarathne, T., Kostle, P. A., Thorne, P. S., O'Shaughnessy, P. T., and Stone, E. A.: Urban enhancement of $PM_{10}$ bioaerosol tracers relative to background locations in the midwestern United States, J. Geophys. Res. Atmos., 121(9), 5071–5089, doi:10.1002/2015JD024538, 2016.

Samaké, A.; Uzu, G.; Martins, J.M.F.; Calas, A.; Vince, E.; Parat, S.; and Jaffrezo, J.L. The unexpected role of bioaerosols in the Oxidative Potential of PM. Sci. Rep., 7, 10978, doi:10.1038/s41598-017-11178-0, 2017.

Samaké, A., Jaffrezo, J.-L., Favez, O., Weber, S., Jacob, V., Canete, T., Albinet, A., Charron, A., Riffault, V., Perdrix, E., Waked, A., Golly, B., Salameh, D., Chevrier, F., Oliveira, D. M., Besombes, J.-L., Martins, J. M. F., Bonnaire, N., Conil, S., Guillaud, G., Mesbah, B., Rocq, B., Robic, P.-Y., Hulin, A., Le Meur, S., Descheemaecker, M., Chretien, E., Marchand, N., and Uzu, G.: Arabitol, mannitol, and glucose as tracers of primary biogenic organic aerosol: the influence of environmental factors on ambient air concentrations and spatial distribution over France, Atmos. Chem. Phys., 19(16), 11013–11030, doi:10.5194/acp-19-11013-2019, 2019a.

Samaké, A., Jaffrezo, J.-L., Favez, O., Weber, S., Jacob, V., Albinet, A., Riffault, V., Perdrix, E., Waked, A., Golly, B., Salameh, D., Chevrier, F., Oliveira, D. M., Bonnaire, N., Besombes, J.-L., Martins, J. M. F., Conil, S., Guillaud, G., Mesbah, B., Rocq, B., Robic, P.-Y., Hulin, A., Le Meur, S., Descheemaecker, M., Chretien, E., Marchand, N., and Uzu, G.: Polyols and glucose particulate species as tracers of primary biogenic organic aerosols at 28 French sites, Atmos. Chem. Phys., 19(5), 3357–3374, doi:10.5194/acp-19-3357-2019, 2019b.

Schnell, I. B., Bohmann, K., and Gilbert, M. T. P.: Tag jumps illuminated - reducing sequence-to-sample misidentifications in metabarcoding studies, Mol. Ecol. Resour., 15(6), 1289–1303, doi:10.1111/1755-0998.12402, 2015.

Taberlet, P., Bonin, A., Zinger, L., and Coissac, E.: Environmental DNA: for biodiversity research and monitoring, Oxford University Press, Oxford, New York., DOI:10.1093/oso/9780198767220.001.0001, 2018.

Verma, S. K., Kawamura, K., Chen, J., and Fu, P.: Thirteen years of observations on primary sugars and sugar alcohols over remote Chichijima Island in the western north pacific, Atmos. Chem. Phys., 18(1), 81–101, doi:https://doi.org/10.5194/acp-18-81-2018, 2018.

Waked, A., Favez, O., Alleman, L. Y., Piot, C., Petit, J.-E., Delaunay, T., Verlinden, E., Golly, B., Besombes, J.-L., Jaffrezo, J.-L., and Leoz-Garziandia, E.: Source apportionment of $PM_{10}$ in a north-western Europe regional urban background site (Lens, France) using positive matrix factorization and including primary biogenic emissions, Atmos. Chem. Phys, 14(7), 3325–3346, doi:10.5194/acp-14-3325-2014, 2014.

Weber, S., Uzu, G., Calas, A., Chevrier, F., Besombes, J.-L., Charron, A., Salameh, D., Ježek, I., Močnik, G. and Jaffrezo, J.-L.: An apportionment method for the oxidative potential of atmospheric particulate matter sources: application to a one-year study in Chamonix, France, Atmos. Chem. Phys., 18(13), 9617–9629, doi:10.5194/acp-18-9617-2018, 2018.

Wei, M., Xu, C., Xu, X., Zhu, C., Li, J., and Lv, G.: Characteristics of atmospheric bacterial and fungal communities in $PM_{2.5}$ following biomass burning disturbance in a rural area of North China Plain, Sci. Total Environ., 651, 2727–2739, doi:10.1016/j.scitotenv.2018.09.399, 2019a.

Wei, M., Xu, C., Xu, X., Zhu, C., Li, J., and Lv, G.: Size distribution of bioaerosols from biomass burning emissions: characteristics of bacterial and fungal communities in submicron ($PM_{1.0}$) and fine ($PM_{2.5}$) particles, Ecotoxicol. Environ. Saf., 171, 37–46, doi:10.1016/j.ecoenv.2018.12.026, 2019b.

Weinert, N., Meincke, R., Gottwald, C., Radl, V., Dong, X., Schloter, M., Berg, G., and Smalla, K.: Effects of genetically modified potatoes with increased zeaxanthin content on the abundance and diversity of rhizobacteria with in vitro antagonistic activity do not exceed natural variability among cultivars, Plant Soil, 326(1–2), 437–452, doi:10.1007/s11104-009-0024-z, 2010.

Welsh, D. T.: Ecological significance of compatible solute accumulation by micro-organisms: from single cells to global climate, FEMS Microbiol. Rev., 24(3), 263–290, doi:10.1111/j.1574-6976.2000.tb00542.x, 2000.

Womack, A. M., Artaxo, P. E., Ishida, F. Y., Mueller, R. C., Saleska, S. R., Wiedemann, K. T., Bohannan, B. J. M., and Green, J. L.: Characterization of active and total fungal communities in the atmosphere over the amazon rainforest, Biogeosciences, 12(21), 6337–6349, doi:10.5194/bg-12-6337-2015, 2015.

Xu, C., Wei, M., Chen, J., Zhu, C., Li, J., Lv, G., Xu, X., Zheng, L., Sui, G., Li, W., Chen, B., Wang, W., Zhang, Q., Ding, A. and Mellouki, A.: Fungi diversity in $PM_{2.5}$ and $PM_1$ at the summit of Mt. Tai: abundance, size distribution, and seasonal variation, Atmos. Chem. Phys., 17(18), 11247–11260, doi:10.5194/acp-17-11247-2017, 2017.

Xue, C., Liu, T., Chen, L., Li, W., Liu, I., Kronstad, J. W., Seyfang, A. and Heitman, J.: Role of an expanded
inositol transporter repertoire in cryptococcus neoformans sexual reproduction and virulence, 1(1),
doi:10.1128/mBio.00084-10, 2010.
Yadav, A. N., Verma, P., Sachan, S. G., Kaushik, R., and Saxena, A. K.: Psychrotrophic microbiomes: molecular
diversity and beneficial role in plant growth promotion and soil health, in Microorganisms for Green Revolution,
vol. 7, pp. 197–240, Springer Singapore, Singapore., doi: 10.1007/978-981-10-7146-1-11, 2018.
Yang, Y., Chan, C., Tao, J., Lin, M., Engling, G., Zhang, Z., Zhang, T., and Su, L.: Observation of elevated fungal
tracers due to biomass burning in the Sichuan Basin at Chengdu City, China, Sci. Total Environ., 431, 68–77,
923 2012.

Yan, C., Sullivan, A. P., Cheng, Y., Zheng, M., Zhang, Y., Zhu, T., and Collett, J. L.: Characterization of
saccharides and associated usage in determining biogenic and biomass burning aerosols in atmospheric fine
particulate matter in the North China Plain, Sci. Total Environ., 650, 2939–2950,
doi:10.1016/j.scitotenv.2018.09.325, 2019.
Yu, X., Wang, Z., Zhang, M., Kuhn, U., Xie, Z., Cheng, Y., Pöschl, U., and Su, H.: Ambient measurement of
fluorescent aerosol particles with a WIBS in the Yangtze River Delta of China: potential impacts of combustion-
related aerosol particles, Atmos. Chem. Phys., 16(17), 11337–11348, doi:10.5194/acp-16-11337-2016, 2016.
Zhu, C., Kawamura, K., and Kunwar, B.: Organic tracers of primary biological aerosol particles at subtropical
Okinawa Island in the western North Pacific Rim: organic biomarkers in the north pacific, Journal of Geophysical
Research: Atmospheres, 120(11), 5504–5523, 2015.