# Peer review of "High levels of primary biogenic organic aerosols are driven by only a few plant-associated microbial taxa"

_Atmospheric Chemistry and Physics, 2019_

## Referee Comment (RC1) · Anonymous Referee #1 · 29 Jan 2020

This manuscript presents a study on the connection between organic tracers and microbial taxa and it also points to the important contribution of leaves from surrounding vegetations. The manuscript is well prepared and the results and findings are very interesting and of importance for the fellow researchers. It could be accepted for publication in the journal.

(1) The title can be shortened or modified to make it concise and highlighted.

(2) L12-13: The specific taxa of bacterials and fungi as the major contributors of PBOA

could be highlighted in the Abstract.

(3) As mentioned in Section 2.1, the meteorological data were collected during the campaign. How about the influence of these environmental factors such as temperature, humidity and so on which were suggested to be essential for the release of primary biological aerosols?
* * *

---

## Referee Comment (RC2) · Anonymous Referee #2 · 17 Feb 2020

The manuscript describes the contribution of primary biogenic organic aerosols (PBOAs) to PM10, which were collected during the summer of 2017 (June–August) in a rural area of France. The quartz fiber filters (24-hour samples) were collected using high-volume sampling systems. The collected samples were analyzed for detailed chemical composition (inorganic ions, OCs and ECs, sugars and sugar alcohols) and for biological constituencies (DNA sequencing and analysis). Soil and vegetation samples were also collected and analyzed with the same techniques for comparison.

The goal of this study was to investigate the association between the chemical composition of PM10 (especially sugar compounds or SCs) and the identified PBOAs. This study is scientifically very important because very little is known about the contribution of bioaerosols to atmospheric particulate matter and what kinds of markers can be used for the quantitative analysis of bioaerosols in particular fungi and bacteria. The manuscript is well written and organized. I have several major comments.

Major Comments:

1. The title of the manuscript doesn't represent the research of this paper (it shows one of the results, but not the overall scope of this study).

2. The author performed a comparison between SC concentrations and collected bioaerosols, assuming that SCs in the atmospheric aerosols are mainly due to PBOAs. Other sources (e.g., biomass burning, the ocean) can also emit SCs (sugars and sugar alcohols). The author is missing the entire discussion of these possible sources. Therefore, without a proper comparison of SC emissions from different sources, the statement regarding "suitable markers" (line 77) should be carefully used. A discussion on other SC sources is needed.

3. Pollen can be a huge contributor to atmospheric PBOAs. Why were only bacteria and fungi collected and analyzed?

4. What standard deviations represented in this paper (e.g., lines 270–282)? It is unclear how they were calculated.

5. Figure 5 is not readable.

6. In lines 148–149, based on which factors (literature data etc.) was the OM/OC conversion factor of 1.8 used? This choice has to be well explained.

Some (not all) minor comments:

Line 16. It is should be "on rural area of France"

Lines 37. References are missing.

Line 119. Why PM10 cut was selected for sampling? Some PBOA have a larger size.

Line 122. How the collected filters were stored prior analyses? (It has to be added to the experimental section).

Lines 143, 170, 172. Company's city (state, country) is missing.

Line 207 (and everywhere in the text). Words "bacteria" and "fungi" should not be capitalized.

Figure 6. Explain what black diamonds represent in this figure.

Line 439. The space should be removed before "."

Line 445. Use OM instead of "organic matter".

Line 477. Use SC not "sugar compounds"

Line 518. Remove extra "."

Line 531. Use "strongly" instead of "highly".

I recommend this manuscript for publication after the author addresses the major questions.

---

## Author Comment (AC1) · 11 Mar 2020

This manuscript presents a study on the connection between organic tracers and microbial taxa and it also points to the important contribution of leaves from surrounding vegetations. The manuscript is well prepared and the results and findings are very interesting and of importance for the fellow researchers. It could be accepted for publication in the journal.

We thank the reviewer for his/her constructive comments that helped to improve the

quality of this work. We have reviewed the comments and made a point by point revision of the article. Detailed responses to the comments are given below, point by point, in blue, including changes made directly to the manuscript, in red (see the main text).

Specific comments:

(1) The title can be shortened or modified to make it concise and highlighted.

We agree with the referee and have changed the title as follows "High levels of primary biogenic organic aerosols are driven by only a few plant-associated microbial taxa" to concisely represent the findings of our research paper.

(2) L12-13: The specific taxa of bacterial and fungi as the major contributors of PBOA could be highlighted in the Abstract.

Thank you for suggestion, this information has been added in the Abstract (see lines 13-16).

(3) As mentioned in Section 2.1, the meteorological data were collected during the campaign. How about the influence of these environmental factors such as temperature, humidity and so on which were suggested to be essential for the release of primary biological aerosols?

We do agree with the referee that some specific environmental factors have been widely suggested as potentially influencing the initial release of primary biological aerosols and their subsequent dispersion in air (Jones and Harrison, 2004; Zhang et al., 2010). However, in our opinion the detailed analysis of these relationships was beyond the scope of this study for two main reasons:

First, we have recently conducted a large study on the spatial behavior of primary sugar compounds (SC) and the identification of their major effective environmental drivers. In this recent study, based on daily (24h) data series covering 16 national sites throughout France, it was clearly demonstrated that the main drivers of daily atmospheric concentrations of SC are ambient air temperature, relative humidity, wind speed, vegetation

density or harvesting activities for sites influenced by agriculture (Samaké et al., 2019).

Second, it has already been shown that bioaerosol concentrations (most likely Bacteria and Fungi) in the near surface atmosphere increase during rainstorms, with some studies also showing that Ascomycota concentrations increase sharply during and immediately after rainstorms (Elbert et al., 2007; Womack et al., 2015). This implies that their initial release is closely linked to the short-term dynamics (i.e. within few hours) of specific environmental factors. Here, we would like to draw the attention of the reviewers to the fact that, in the present study, some consecutive quartz filter samples (maximum 2 days) with low OM concentrations have been pooled for high throughput DNA sequencing. The resultant composite data at a temporal resolution of 2-days are too coarse to allow a realistic study of the environmental factors that effectively drive the short-term dynamics of primary biological aerosol concentrations.

Thus, extending the correlation analysis between local meteorology (few hours) and the relative abundances of fungal sequences at the genus level (aggregated over 2 consecutive days) would not, in our opinion, provide valuable additional information.

References

Elbert, W., Taylor, P. E., Andreae, M. O., and Pöschl, U.: Contribution of fungi to primary biogenic aerosols in the atmosphere: wet and dry discharged spores, carbohydrates, and inorganic ions, Atmos. Chem. Phys., 7(17), 4569–4588, doi:10.5194/acp-7-4569-2007, 2007.

Jones, A. M. and Harrison, R. M.: The effects of meteorological factors on atmospheric bioaerosol concentrations—a review, Sci. Environ." 326(1), 151–180, doi:10.1016/j.scitotenv.2003.11.021, 2004.

Samaké, A., Jaffrezo, J.-L., Favez, O., Weber, S., Jacob, V., Canete, T., Albinet, A., Charron, A., Riffault, V., Perdrix, E., Waked, A., Golly, B., Salameh, D., Chevrier, F., Oliveira, D. M., Besombes, J.-L., Martins, J. M. F., Bonnaire, N., Conil, S., Guillaud,

G., Mesbah, B., Rocq, B., Robic, P.-Y., Hulin, A., Le Meur, S., Descheemaecker, M., Chretien, E., Marchand, N., and Uzu, G.: Arabitol, mannitol, and glucose as tracers of primary biogenic organic aerosol: the influence of environmental factors on ambient air concentrations and spatial distribution over France, Atmos. Chem. Phys., 19(16), 11013–11030, doi:10.5194/acp-19-11013-2019, 2019.

Womack, A. M., Artaxo, P. E., Ishida, F. Y., Mueller, R. C., Saleska, S. R., Wiedemann, K. T., Bohannan, B. J. M., and Green, J. L.: Characterization of active and total fungal communities in the atmosphere over the Amazon rainforest, Biogeosciences, 12(21), 6337–6349, doi:10.5194/bg-12-6337-2015, 2015.

Zhang, T., Engling, G., Chan, C.-Y., Zhang, Y.-N., Zhang, Z.-S., Lin, M., Sang, X.-F., Li, Y. D., and Li, Y.-S.: Contribution of fungal spores to particulate matter in a tropical rainforest, Environmental Research Letters, 5(2), 024010, doi:10.1088/1748-9326/5/2/024010, 2010.

---

## Author Comment (AC2) · 11 Mar 2020

The manuscript describes the contribution of primary biogenic organic aerosols (PBOAs) to PM10, which were collected during the summer of 2017 (June–August) in a rural area of France. The quartz fiber filters (24-hour samples) were collected using high-volume sampling systems. The collected samples were analyzed for detailed chemical composition (inorganic ions, OCs and ECs, sugars and sugar alcohols) and for biological constituencies (DNA sequencing and analysis). Soil and vegetation samples were also collected and analyzed with the same techniques for comparison. The

goal of this study was to investigate the association between the chemical composition of PM10 (especially sugar compounds or SCs) and the identified PBOAs. This study is scientifically very important because very little is known about the contribution of bioaerosols to atmospheric particulate matter and what kinds of markers can be used for the quantitative analysis of bioaerosols in particular fungi and bacteria. The manuscript is well written and organized. I have several major comments.

We thank the anonymous referee for taking the time to evaluate this manuscript, and for all the suggestions for modifications and comments that helped us improve the quality of this work. We have taken all the comments into account and have made a point by point revision. Detailed responses to the comments are given below, point by point, in blue, including changes made directly to the manuscript, in red.

Major Comments:

(1) The title of the manuscript doesn't represent the research of this paper (it shows some of the results, but not the overall scope of this study)

We thank the reviewer for this remark, which was also suggested by anonymous referee 1. The title has been changed as follows "High levels of primary biogenic organic aerosols are driven by only a few plant-associated microbial taxa" in the main text.

(2) The author performed a comparison between SC concentrations and collected bioaerosols, assuming that SCs in the atmospheric aerosols are mainly due to PBOAs. Other sources (e.g., biomass burning, the ocean) can also emit SCs (sugars and sugar alcohols). The author is missing the entire discussion of these possible sources. Therefore, without a proper comparison of SC emissions from different sources, the statement regarding "suitable markers" (line 77) should be carefully used. A discussion on other SC sources is needed.

The reviewer is correct that other sources, including biomass burning and the ocean, have sometimes been proposed as potential emitters of SCs (Yang et al., 2012). However, recent studies conducted at several sites across France have shown a weak correlation between daily concentrations of SC and levoglucosan in PM2.5 and PM10 collected throughout the year (Golly et al., 2018; Samaké et al., 2019a). In the present study, there was no significant correlation between primary sugar species and levoglucosan, a tracer of biomass burning, in our PM10 time series. In addition, primary sugar compounds were not significantly related to two typical marine ions (e.g. Na+ or Cl−) or methanesulfonic acid, a tracer of marine biogenic activity (Zhu et al., 2015). It therefore seems unlikely that sources of SC in PM10 from biomass burning or ocean were significant at this site.

As suggested by the reviewer, a discussion of other potential sources proposed in a few previous studies has been added in the main text as follows:

Lines 68-73: "SC species are emitted from biologically derived sources (Medeiros et al., 2006, Verma et al., 2018) and have sometimes been detected in aerosols taken from air masses influenced by smoke from biomass burning (Fu et al., 2012; Yang et al., 2012). However, recent studies conducted at several sites across France revealed a weak correlation between daily concentrations of SC and levoglucosan in PM2.5 and PM10 collected throughout the year (Golly et al., 2018; Samaké et al., 2019a). This suggests that open burning of biomass is not a significant source of SC in the environments studied here".

Lines 490-497: "However, in our study, SC species are not correlated (R = −0.09, p = 0.46; Fig. S7) with levoglucosan during the campaign period, confirming that biomass burning is not an important source of airborne microbial taxa associated with SCs in our PM10 series. Bubble bursting associated with sea spray could also potentially be a source of Bacteria, Fungi and water-soluble organic species, along with sea salts, to PM10 (Prather et al., 2013; Zhu et al., 2015). However, SC species were not found to be significantly related to Cl− (R = −0.14, p = 0.28) or Na+ (R = −0.18, p = 0.16), which are two inorganic tracers typical of marine sources; nor correlated with methanesulfonic acid (R = −0.05, p = 0.69), a well-known tracer of biogenic marine activity (Arndt et al.,

2017; Gaston et al., 2010). It therefore seems unlikely that the sources of SCs from marine environments were significant at this site".

(3) Pollen can be a huge contributor to atmospheric PBOAs. Why were only bacteria and fungi collected and analyzed?

The reviewer is correct that pollen can be a significant contributor to atmospheric PBOAs. However, this abundance is expected to also depend on the PM size range considered. Individual airborne pollen grains generally range about 10-100 $\mu$m (Manninen et al., 2014; Yoo et al., 2017), while fungal spores are much smaller, 1–30 $\mu$m, and most often < 10 $\mu$m (Després et al., 2012; Manninen et al., 2014). Similarly, the diameter of airborne Bacteria is generally between 0.25 and about 8 $\mu$m (Yoo et al., 2017). Our study therefore focused on Bacteria and Fungi as they are generally the dominant biological component of ambient aerosols in the size range of 2–10 $\mu$m (Zhang et al., 2010), discussed in this study.

(4) What standard deviations represented in this paper (e.g., lines 270–282)? It is unclear how they were calculated.

The standard deviations presented in this section measure the amount for dispersion of each SC species measured relative to its mean value. They represent SD standard deviations. This is now clarified in the main text.

(5) Figure 5 is not readable.

We are a bit surprised by this comment as heatmaps are commonly used in microbiology studies to facilitate the visual presentation and exploration of complex correlation patterns. If the comment is about the quality of the figure, we will provide the reviewer with a new figure with a much higher resolution for final publication.

(6) In lines 148–149, based on which factors (literature data etc.) was the OM/OC conversion factor of 1.8 used? This choice has to be well explained.

This value of 1.8 for the OM/OC ratio was chosen on the basis of previous studies

carried out in France, for the purpose of spatial comparison. In a recent study, we performed a mass balance between PM10 chemistry and TEOM measurements where the conversion factor of 1.8 was found to be consistent with a correct reconstruction of the PM10 mass (Favez et al., 2010; Golly et al., 2018; Petit et al., 2015).

Our choice, as suggested by the reviewer, is now explained in the main text as follows:

Lines 158-159 : "This value of 1.8 for the OM/OC ratio was chosen on the basis of previous studies carried out in France (Samaké et al., 2019b, and reference therein)".

Some (not all) minor comments:

(7) Line 16. It is should be "on rural area of France"

This has been changed in the main text (see line 16).

(8) Lines 37. References are missing.

Missing references have been added in the main text (see line 40-41).

(9) Line 119. Why PM10 cut was selected for sampling? Some PBOA have a larger size.

In European countries, including France, health alerts on particulate matter are based on measurements of particles less than 10 $\mu$m in diameter. More details on European Union particulate matter standards can be found elsewhere (Priemus and Schutte-Postma, 2009). Therefore, our study focused on understanding the PBOA in the PM10 fraction.

(10) Line 122. How the collected filters were stored prior analyses? (It has to be added to the experimental section).

This information has been added (see lines 131-132)

(11) Lines 143, 170, 172. Company's city (state, country) is missing.

This information has been added (see lines 155, 183, 185)

(12) Line 207 (and everywhere in the text). Words "bacteria" and "fungi" should not be capitalized

We thank the reviewer for this suggestion. However, according to the conventions of microorganisms nomenclature, the name of the microbiome phylum should generally begin with a capital letter.

(13) Figure 6. Explain what black diamonds represent in this figure.

Thank you for this suggestion. A detailed explanation has now been added (see line 432-434).

(14) Line 439. The space should be removed before "."

This extra space has been removed.

(15) Line 445. Use OM instead of "organic matter".

This has been changed in the main text.

(16) Line 477. Use SC not "sugar compounds"

This has been changed in the main text.

(17) Line 518. Remove extra "."

This has been changed in the main text.

(18) Line 531. Use "strongly" instead of "highly".

This has been changed in the main text.

References

Després, V. R., Alex Huffman, J., Burrows, S. M., Hoose, C., Safatov, A. S., Buryak, G., Fröhlich-Nowoisky, J., Elbert, W., Andreae, M. O., Pöschl, U., and Jaenicke, R.: Primary biological aerosol particles in the atmosphere: a review, Tellus B: Chemical and Physical Meteorology, 64(1), 15598, doi:10.3402/ tellusb.v64i0.15598, 2012.

Favez, O., Haddad, I. E., Piot, C., Boreave, A., Abidi, E., and Marchand, N.: Inter-comparison of source apportionment models for the estimation of wood burning aerosols during wintertime in an Alpine city (Grenoble, France), Atmos. Chem. Phys., 20, 2010.

Golly, B., Waked, A., Weber, S., Samaké, A., Jacob, V., Conil, S., Rangognio, J., Chrétien, E., Vagnot, M.-P., Robic, P.-Y., Besombes, J.-L. and Jaffrezo, J.-L.: Organic markers and OC source apportionment for seasonal variations of PM2.5 at 5 rural sites in France, Atmos. Environ., 198, 142–157, doi:10.1016/j.atmosenv.2018.10.027, 2018.

Manninen, H. E., Bäck, J., Sihto-Nissilä, S.-L., Huffman, J. A., Pessi, A.-M., Hiltunen, V., Aalto, P. P., Hidalgo Fernández, P. J., Hari, P., Saarto, A., Kulmala, M., and Petäjä, T.: Patterns in airborne pollen and other primary biological aerosol particles (PBAP), and their contribution to aerosol mass and number in a boreal forest, Boreal Environ. Res., 383–405, doi:hdl.handle.net/10138/165208, 2014.

Petit, J.-E., Favez, O., Sciare, J., Crenn, V., Sarda-Estève, R., Bonnaire, N., Močnik, G., Dupont, J.-C., Haeffelin, M., and Leoz-Garziandia, E.: Two years of near real-time chemical composition of submicron aerosols in the region of Paris using an aerosol chemical speciation monitor (ACSM) and a multi-wavelength Aethalometer, Atmos. Chem. and Phys., 15(6), 2985–3005, doi:10.5194/acp-15-2985-2015, 2015.

Priemus, H. and Schutte-Postma, E.: Notes on the Particulate Matter Standards in the European Union and the Netherlands, IJERPH, 6(3), 1155–1173, doi:10.3390/ijerph6031155, 2009.

Samaké, A., Jaffrezo, J.-L., Favez, O., Weber, S., Jacob, V., Canete, T., Albinet, A., Charron, A., Riffault, V., Perdrix, E., Waked, A., Golly, B., Salameh, D., Chevrier, F., Oliveira, D. M., Besombes, J.-L., Martins, J. M. F., Bonnaire, N., Conil, S., Guillaud, G., Mesbah, B., Rocq, B., Robic, P.-Y., Hulin, A., Le Meur, S., Descheemaecker, M., Chretien, E., Marchand, N., and Uzu, G.: Arabitol, mannitol, and glucose as tracers of primary biogenic organic aerosol: the influence of environmental factors on ambient

air concentrations and spatial distribution over France, Atmos. Chem. Phys., 19(16), 11013–11030, doi:10.5194/acp-19-11013-2019, 2019a.

Samaké, A., Jaffrezo, J.-L., Favez, O., Weber, S., Jacob, V., Albinet, A., Riffault, V., Perdrix, E., Waked, A., Golly, B., Salameh, D., Chevrier, F., Oliveira, D. M., Bonnaire, N., Besombes, J.-L., Martins, J. M. F., Conil, S., Guillaud, G., Mesbah, B., Rocq, B., Robic, P.-Y., Hulin, A., Le Meur, S., Descheemaecker, M., Chretien, E., Marchand, N., and Uzu, G.: Polyols and glucose particulate species as tracers of primary biogenic organic aerosols at 28 French sites, Atmos. Chem. Phys., 19(5), 3357–3374, doi:10.5194/acp-19-3357-2019, 2019b.

Yang, Y., Chan, C., Tao, J., Lin, M., Engling, G., Zhang, Z., Zhang, T., and Su, L.: Observation of elevated fungal tracers due to biomass burning in the Sichuan Basin at Chengdu City, China, Sci. Tot. Environ, 431, 68–77, 2012.

Yoo, K., Lee, T. K., Choi, E. J., Yang, J., Shukla, S. K., Hwang, S., and Park, J.: Molecular approaches for the detection and monitoring of microbial communities in bioaerosols: A review, Journal of Environmental Sciences, 51, 234–247, doi:10.1016/j.jes.2016.07.002, 2017.

Zhang, T., Engling, G., Chan, C.-Y., Zhang, Y.-N., Zhang, Z.-S., Lin, M., Sang, X.-F., Li, Y. D., and Li, Y.-S.: Contribution of fungal spores to particulate matter in a tropical rainforest, Environmental Research Letters, 5(2), 024010, doi:10.1088/1748-9326/5/2/024010, 2010.

---

## Author Response (AR1)

Institut des Géosciences de l'Environnement

Dr Jean Martins (jean.martins@univ-grenoble-alpes.fr )

Dr Abdoulaye Samaké (abdoulaye.samake2@univ-grenoble-alpes.fr )

460, Rue de la Piscine

BP 53, Domaine Universitaire

38041 Grenoble, France

[Figure]

Grenoble, March 11ˢᵗ, 2020.

Dear Prof Alex Huffman,

Thank you for your consideration, please find enclosed our revised manuscript on "High levels of primary biogenic organic aerosols in the atmosphere in summer are driven by only a few microbial taxa from the leaves of surrounding plants" by Abdoulaye Samaké et al. (MS No.: acp-2019-1147).

We would like to thank very much referees for their constructive comments and suggestions. We have studied the comments of both referee #1 and #2. We made revisions point by point in the attached file.

[revised manuscript text omitted]